# Radiative Asymptotic Symmetries of 3D Einstein–Maxwell Theory

Jorrit Bosma[1], Marc Geiller[2], Sucheta Majumdar[2], Blagoje Oblak[3,4]

[1] *Institute for Theoretical Physics, ETH Zürich, CH 8093 Zürich, Switzerland*
[2] *ENS de Lyon, CNRS, Laboratoire de Physique, F-69342 Lyon, France*
[3] *CPHT, CNRS, Ecole Polytechnique, IP Paris, F-91128 Palaiseau, France*
[4] *Optique Non-linéaire Théorique, Université Libre de Bruxelles,*
*Campus Plaine C.P. 231, B-1050 Bruxelles, Belgium*

**Abstract**

We study the null asymptotic structure of Einstein–Maxwell theory in three-dimensional (3D) spacetimes. Although devoid of bulk gravitational degrees of freedom, the system admits a massless photon and can therefore accommodate electromagnetic radiation. We derive fall-off conditions for the Maxwell field that contain both Coulombic and radiative modes with non-vanishing news. The latter produces non-integrability and fluxes in the asymptotic surface charges, and gives rise to a non-trivial 3D Bondi mass loss formula. The resulting solution space is thus analogous to a dimensional reduction of 4D pure gravity, with the role of gravitational radiation played by its electromagnetic cousin. We use this simplified setup to investigate choices of charge brackets in detail, and compute in particular the recently introduced Koszul bracket. When the latter is applied to Wald–Zoupas charges, which are conserved in the absence of news, it leads to the field-dependent central extension found earlier in [1]. We also consider (Anti-)de Sitter asymptotics to further exhibit the analogy between this model and 4D gravity with leaky boundary conditions.

# 1    Introduction and summary

Despite its key role in physics, the behaviour of quantum gauge field theories at large distances and low energies remains poorly understood. Indeed, the bulk degrees of freedom of gauge theories can be quantized with standard Hamiltonian methods [2], but their physical boundary states are exceedingly sensitive to choices of fall-off conditions and thus require a separate treatment. This subtlety is especially striking in cases that contain propagating modes (as opposed to topological field theories), where the global symmetries of boundary states—*i.e.* asymptotic symmetries—suffer from radiative ambiguities. The present work is therefore devoted to a detailed classical investigation of such issues in a simple gauge system: gravitation coupled to electrodynamics in three spacetime dimensions (3D). In this introduction, we first motivate the subject, then present an overview of our main results along with a plan of the paper.

**Motivations.**    The difficulties in quantizing the infrared sector of gauge theories are best exemplified by gravitation, whose general covariance requires that conserved quantities (say energy or momentum) be defined in terms of asymptotic fluxes as opposed to local current densities [3, 4]. This is an old, general and standard result whose counterpart for 4D gravitational radiation involves the infinite-dimensional Bondi–Metzner–Sachs (BMS) group at null infinity, first introduced half a century ago [5–11]. Such constructions are indeed well known by now, but the full scope of their implications has only recently become apparent thanks to new insights on holography in Minkowskian spacetimes [12–15] and the discovery of deep links between asymptotic symmetries and the infrared sector of gauge theories [16–19]. Following these ideas, asymptotic symmetries were studied in numerous contexts, notably including electrodynamics and Yang–Mills theory [20–25]. A common aspect of all these works is that one's choice of fall-off conditions for field components determines the phase space of available field configurations, which in turn influences the soft sector of the theory and the resulting symmetries.

The present paper is devoted to the asymptotic symmetries of 3D gravity coupled to a Maxwell field. This setup was first studied in [1], but we shall argue that a crucial sector of radiative field configurations was overlooked in that reference. Indeed, our investigation is precisely motivated by the (much more involved) issue of radiative ambiguities in the asymptotic symmetries of 4D general relativity. In that case, the presence of gravitational waves implies that the radiative phase space at null infinity is an open system [26–28], which has important technical and conceptual implications. The first is that asymptotic charges are not conserved in time but obey instead flux-balance relations, such as the Bondi mass loss relating the decrease of mass to the radiation crossing null infinity. The second is that charges are generally non-integrable, as the associated symmetries fail to be Hamiltonian due to the presence of symplectic fluxes. This hinders the study of the corresponding asymptotic charge algebras, since no canonical bracket can *a priori* be defined.

In fact, a tentative solution to this puzzle was put forward by Barnich and Troessaert [12–14], yielding *e.g.* an asymptotic charge algebra extended by a suggestive field-dependent cocycle [14]. But the prescription of [14] is ultimately ambiguous, since it relies on an arbitrary splitting between integrable terms and non-integrable fluxes when defining the very notion of 'charge'. It is therefore

essential to elucidate the origin and the extent of this arbitrariness, the goal being to eventually isolate unambiguous boundary symmetries. In the specific case of reference [14], the split ambiguity can actually be fixed by the Wald–Zoupas (WZ) prescription [29–31] that singles out the integrable charge by requiring conservation in the absence of radiation. However, it is unclear if an analogous way out exists in general radiative cases. Radiative asymptotic symmetries are thus crucial for quantum gravity and gauge theories as a whole, and for Minkowskian holography in particular [13, 32–39]. Note that similar questions can also be raised in (Anti)-de Sitter [(A)dS] spacetimes, where radiation and holography with porous boundary conditions were recently studied in [40–57].

Our focus on a 3D toy model is motivated by the desire to simplify matters, while still retaining the essential features of realistic radiative systems. Indeed, pure 3D gravity is a topological field theory (no propagating degrees of freedom) that has often served as a fruitful testbed for asymptotic symmetries, starting with the seminal work [58] on the boundary Virasoro algebra of $\text{AdS}_3$. The analogous symmetry of 3D Minkowskian spacetimes is the $\text{BMS}_3$ algebra [59–61]. In both AdS and flat cases, the absence of radiation allows one to delve deep into holographic territory on a purely group-theoretic basis. For instance, unitary representations of asymptotic symmetries [62–68] and their relation to bulk metrics through coadjoint orbits and geometric actions [69–78] are well established in both situations, as is the Cardy-ology of black holes and cosmological solutions [79–81]. No such control is available in the 4D realm, where the study of coadjoint orbits [82] and geometric actions [83, 84] is in its infancy, and mostly limited to the sector without radiation.

Hence our interest in the 3D Einstein–Maxwell system: it is radiative (the electromagnetic field has one local degree of freedom) but sits between 3D and 4D general relativity in terms of complexity, as it shares the geometrical simplicity of the former while also describing radiative aspects relevant to the latter. Its symmetries were studied in the aforementioned reference [1] (see also [85, 86]), where the authors identified a non-integrable contribution to the charges, which they interpreted as being sourced by electromagnetic 'news'. However, this interpretation should be done with care, as a slightly more general setup reveals a key subtlety. This is actually the gist of our work: we generalize the analysis of [1] by considering weaker fall-offs for the Maxwell field. In particular, while the fall-offs of [1] include logarithmic terms (as required for the 3D Coulomb solution) and integer powers of the radius $r$, here we also allow for terms in half-integer powers of $r$. This is because radiative fall-offs for the 3D Maxwell field in (retarded) Bondi coordinates $(r, u, \phi)$ and in radial gauge take the form (see fig. 1)

$$A_\phi = C\sqrt{r} + \dots, \qquad A_u = E(\ln r) + G + \dots. \qquad (1.1)$$

Here $C(u, \phi)$ is an unconstrained function on $\mathscr{I}^+$ and denotes the radiative electromagnetic data, $E(\phi)$ is the electric charge aspect, $G(u, \phi)$ is another unconstrained function, and the dots are subleading terms (which we will analyse in detail). With such relaxed fall-offs, 3D Einstein–Maxwell theory with electromagnetic radiation becomes analogous to a dimensional reduction of 4D pure gravity with gravitational radiation [59, 87]. In particular, both the symplectic structure and the charges at null infinity involve the variational one-form $C\delta N$, where

$$N(u, \phi) := \partial_u C(u, \phi) := \dot{C}(u, \phi) \qquad (1.2)$$

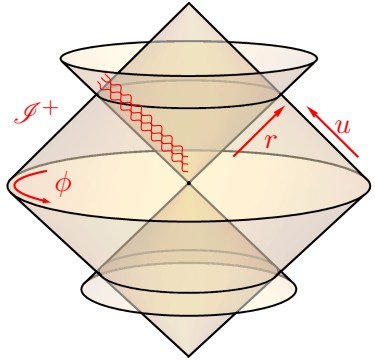

Figure 1: A Penrose diagram of 3D Minkowski spacetime, showing retarded Bondi coordinates $(r, u, \phi)$ along with light-cones centred at the origin. Future null infinity is the region denoted $\mathscr{I}^+$, where $r \to \infty$ at finite $(u, \phi)$. The presence of electromagnetic radiation (wiggly lines) reaching $\mathscr{I}^+$ is diagnosed by a non-zero news function (1.2). Adapted from [88, fig. 1].

is the electromagnetic news. For comparison, recall that the news in 4D gravity is a symmetric tracefree rank-2 tensor $N_{ab}$, while in 4D Maxwell theory it is a vector $N_a$ in terms of coordinates $x^a$ on the celestial 2-sphere:

|  | 4D | 3D |
| --- | --- | --- |
| Einstein | $N_{ab}$ | $\emptyset$ |
| Maxwell | $N_a$ | $N$ |

This highlights again that 3D Maxwell exhibits the simplest form of news. Coupling it to Einstein gravity makes it possible to study the interplay between this news and gravitational asymptotic symmetries. In particular, the Bondi mass aspect $M$ then satisfies a mass loss formula $\dot{M} = -2N^2$, analogous to its celebrated 4D version with gravitational news replaced by its electromagnetic counterpart. No such structure occurs in the analysis of [1], due to stronger fall-offs setting $C = 0$.

As stated above, the new boundary conditions presented here provide an ideal arena to study all the subtleties related to radiation, such as non-integrability, flux-balance laws and ambiguities in one's choice of charge brackets. An additional goal is to contribute to the study of asymptotic symmetries for general relativity coupled to matter fields. For 4D Einstein–Maxwell theory, asymptotic symmetries were already partly studied in [89, 90], where the first reference focusses on WZ charges. Given our observations below on charge algebras and their field-dependent cocycles, it would be interesting to also study the charge algebra in the 4D case. We hope to come back to this in the future.

**Outline and summary of results.** The paper is organized as follows. We start in section 2 by analysing pure Maxwell theory in 3D Minkowski spacetime. From the electromagnetic field produced by a compact source, we deduce the fall-offs to be used later in Einstein–Maxwell theory in Bondi coordinates. We then compute the 3D electromagnetic memory effect, which consists of a net change of angular velocity caused by radiation on any charged test mass near null infinity. In contrast to the 4D case [91], this is not directly related to a change of U(1) asymptotic charges; the mismatch ultimately stems from the difference between Coulombic and radiative falls-off in any spacetime dimension other than four.

Section 3 is devoted to the solution space of Einstein–Maxwell theory with vanishing cosmological constant. We explain there how the field equations are solved following the Bondi hierarchy, deferring the details to appendix A. The upshot is that the solution space contains, among others, two functions $M(u, \phi)$ and $L(u, \phi)$. These are respectively the mass and angular momentum aspects in the metric, subject to the evolution equations (3.10), the first of which is the Bondi mass loss sourced by the electromagnetic news (1.2). In addition, the solution space contains two completely unconstrained functions on $\mathscr{I}^+$, namely the electromagnetic shear $C(u, \phi)$ and a function $G(u, \phi)$ as in (1.1). Only $G$ was present in [1], where it gave rise to non-integrable asymptotic charges and was thus identified as 'news'. We will show instead that $G(u, \phi)$ should be considered as sourcing a 'spurious flux', while the 'true news' is the quantity $\dot{C}$ sourcing the Bondi mass loss through a flux of energy momentum at $\mathscr{I}^+$.

Note that the phase space thus obtained contains *two* free functions on $\mathscr{I}^+$, namely $C(u, \phi)$ and $G(u, \phi)$. This may seem perplexing given that the theory admits a single degree of freedom. Actually, an analogous situations occurs in 4D pure gravity, where one can relax boundary conditions so that the solution space contains other undetermined functions of $u$ on top of the two degrees of freedom in the asymptotic shear $C_{ab}$. This includes for instance the induced boundary metric on $\mathscr{I}^+$ [53, 54] or the trace of $C_{ab}$ in Newman–Unti gauge [54, 92], and it allows one to describe physically relevant solutions with $C_{ab} = 0$, such as Robinson–Trautman spacetimes [93–95]. The metric on $\mathscr{I}^+$ can similarly be relaxed in 3D pure gravity, resulting in a phase space with *three* unspecified functions of $u$ even though the bulk theory has no propagating degrees of freedom at all [96, 97]. In the case of 3D Einstein–Maxwell theory, the known radiative configurations are such that $C \neq 0$ and $G = 0$; in fact, $G$ may be seen as a gauge redundancy. It is nevertheless possible that the theory admits interesting solutions with $C = 0$ but $G \neq 0$, so we will initially keep both fields non-zero in our analysis of symmetries and charges.

In section 4, we turn to the characterization of asymptotic symmetries. These are labelled by an asymptotic Killing vector field $\xi$ and a U(1) gauge parameter $\varepsilon$ on $\mathscr{I}^+$, spanning a semi-direct sum $\mathfrak{bms}_3 \mathbin{\oplus\mkern-8mu+} C^\infty(\mathscr{I}^+)$. A striking aspect of this construction is that the U(1) parameter at leading order has an arbitrary dependence on retarded time $u$, whose origin is the free $u$-dependent function $G$ in the solution space (1.1). Indeed, one can see from the transformation laws (4.4) that setting $G = 0$ fixes the arbitrary time dependence, reducing the aforementioned algebra to a semi-direct sum $\mathfrak{bms}_3 \mathbin{\oplus\mkern-8mu+} C^\infty(S^1)$. Irrespective of this feature, we find that the (integrands of variations of) asymptotic charges are given by (4.10) and exhibit two non-integrable terms arising from supertranslations. The first such term is $N\delta C$, involving the radiative data of (1.1) and the

news (1.2); the second is $G\,\delta E$, where $E$ is the electric charge aspect in (1.1). These non-integrable contributions imply that a splitting prescription is required for the choice of integrable parts and the bracket of surface charges. This is the (expected) key subtlety anticipated in the motivations above.

Section 5 is therefore devoted to the study of the choice of bracket. We start with the Barnich–Troessaert (BT) prescription [14] in the case $G \neq 0$, whereupon the charge algebra reproduces that of asymptotic symmetries with a field-dependent central extension (really a 2-cocycle) given in (5.6) below. In the case $G = 0$, which must be treated with care since it makes the U(1) gauge parameter field-dependent, we define WZ charges that are conserved when the news (1.2) vanishes. The resulting charge algebra features once again a field-dependent 2-cocycle. Both such extensions were previously found in [1], so they are unaffected by our weaker choice of fall-offs.

The presence of field-dependent cocycles is expected to affect the unitary representations of charge algebras, hence quantum theories having these algebras as symmetries. It is therefore important to ask whether field-dependent cocycles are a generic feature, or if they can be removed with suitable choices of integrable splits or brackets. To this end, we first discuss the so-called Noether split and bracket introduced in [98, 99]; the ensuing algebra closes with a *vanishing* cocycle (even the central extension of [60] is absent!), but integrable charges fail to be conserved in the absence of radiation. We therefore turn to the Koszul bracket[1] introduced in [100], whose key advantage is to be independent of the choice of split between integrable and flux pieces. The Koszul bracket gives a surprising result: when $G \neq 0$, the cocycle in the charge algebra is non-zero but *field-independent*, as in pure 3D gravity [60]; but setting $G = 0$ yields the same field-dependent cocycle as with the BT bracket of WZ charges. Moreover, this last cocycle turns out to be non-zero even in the absence of electromagnetic news, suggesting that field-dependent extensions are a genuine feature of 3D Einstein–Maxwell theory.

In section 6, we switch on a cosmological constant (with arbitrary sign). Einstein–Maxwell theory in AdS$_3$ was previously studied in [86] in Bondi gauge, and in [101, 102] in the Fefferman–Graham gauge that admits no well-defined flat limit. In Bondi gauge, introducing $\Lambda \neq 0$ is straightforward and the flat limit is well-defined [61]. Working in (A)dS$_3$ then deepens the analogy between the present 3D model with electromagnetic radiation and pure 4D gravity with gravitational radiation. For instance, we find that $\Lambda \neq 0$ allows no half-integer powers of $r$ in the fall-offs, analogously to the absence of logarithmic terms in (A)dS$_4$ [53, 54]. Furthermore, when $\Lambda \neq 0$, initial conditions on a cut of $\mathscr{I}^+$ need to be specified for only finitely many components of the Maxwell field, while $\Lambda = 0$ involves an infinite tower of data subject to evolution equations. This is similar to what happens in 4D pure gravity when studying the role of subleading tensors in the angular metric. We finally conclude in section 7 with some perspectives for future work.

---

[1]We thank Adrien Fiorucci for encouraging us to study the Koszul bracket, and for sharing preliminary results based on joint work with Glenn Barnich and Romain Ruzziconi [100].

# 2 Pure Maxwell theory in 3D

This section sets the stage by spelling out basic facts on 3D pure electrodynamics. We first solve Maxwell's equations for a compact source to motivate fall-off conditions in Lorenz gauge, before finding the analogous fall-offs in Bondi gauge. We then briefly comment on electromagnetic memory effects and point out the difference of their behavior with respect to the 4D case.

## 2.1 Electromagnetic field from compact sources

Consider 3D Minkowski spacetime, choose some inertial coordinates $x^\mu = (t, \mathbf{x})$, and let $j_\mu(\mathbf{x}, t)$ be some current density that is compactly supported in space. The current acts as a source for an electromagnetic field $A_\mu$ such that $\partial_\mu F^{\mu\nu} = -j^\nu$, which yields $\Box A_\mu = -j_\mu$ in the Lorenz gauge ($\partial_\mu A^\mu = 0$). Our goal is to find the asymptotics of $A_\mu$ in Bondi coordinates.

The first step is to solve Maxwell's equation $\Box A_\mu = -j_\mu$. Using the (retarded) Green's function of the 3D d'Alembert operator, the solution reads

$$A_\mu(\mathbf{x}, t) = \frac{1}{2\pi} \int d^2\mathbf{y} \int_{-\infty}^{t-|\mathbf{x}-\mathbf{y}|} ds \, \frac{j_\mu(\mathbf{y}, s)}{\sqrt{(t-s)^2 - |\mathbf{x}-\mathbf{y}|^2}}. \tag{2.1}$$

In contrast to the more familiar 4D case, the Green's function that gives (2.1) is *not* localized on a light-cone emanating from the source. This may be viewed as a violation of the Huygens principle in 3D [103], and it ultimately results in (logarithmic) infrared divergences that are absent in 4D. Indeed, consider (2.1) for a point charge. If the latter is static (say at the origin $r = 0$), its current density is $j_0(\mathbf{y}, s) = q\delta^{(2)}(\mathbf{y})$ and $j_1 = j_2 = 0$. Plugging this in (2.1) yields $A_1 = A_2 = 0$, but $A_0$ involves an integral that diverges owing to the infinite lower bound on $s$. The key point, though, is that this divergence is a constant that depends on none of the spacetime coordinates. To see this, fix the radius $r = |\mathbf{x}|$ and replace the lower integration bound in (2.1) by some cutoff time $-T$ such that $T \gg r$ and $T \gg |t - r|$. Then

$$A_0(r, t) = \frac{q}{2\pi} \int_{-T}^{t-r} \frac{ds}{\sqrt{(t-s)^2 - r^2}} = \frac{q}{2\pi} \ln\left(2\frac{T+t}{r}\right) = -\frac{q}{2\pi} \ln(r) + \frac{q}{2\pi} \ln(2T) + \mathcal{O}(T^{-1}) \tag{2.2}$$

in the limit $T \to +\infty$, where the divergent term $\ln(2T)$ is indeed independent of both $r$ and $t$. Thus the Coulombic electrostatic potential in 3D is

$$A_0(r) = -\frac{q}{2\pi} \ln r \tag{2.3}$$

up to an irrelevant additive constant, as was in fact expected from the Green's function of the 2D Laplacian. Note that the same behaviour holds near future null infinity (see fig. 1), where $r \to \infty$ and $t \to \infty$ with $u = t - r$ finite.

In 4D, Coulombic and radiative fall-offs coincide. This is not so in 3D, where the Coulombic potential (2.3) has little to do with its radiative counterpart. One can again verify this from (2.1), now letting $j_\mu$ be the current density of a moving point charge so that $j_\mu(\mathbf{y}, s) = qv_\mu(s)\delta^{(2)}(\mathbf{y}-\mathbf{X}(s))$, where $v_0 := 1$ and $v_i := \partial X_i/\partial s$ in terms of some timelike worldline $\mathbf{X}(s) = (X_1(s), X_2(s))$. For

simplicity, we shall assume that the velocity $\mathbf{v} = (v_1, v_2)$ of the source is small compared with the speed of light. Then the vector potential behaves as

$$A_i(\mathbf{x}, t) \sim \frac{q}{2\pi} \int_{-\infty}^{t-r} \frac{v_i(s)\, \mathrm{d}s}{\sqrt{(t-s)^2 - r^2}} \tag{2.4}$$

at leading order in $|\mathbf{v}|$, and in the limit where $r$ and $t$ are large but $t - r$ is finite. To make further progress, Fourier-transform the velocity as

$$v_j(s) = \int_{-\infty}^{+\infty} \mathrm{d}\omega\, \tilde{v}_j(\omega)\, e^{i\omega s}, \tag{2.5}$$

where $\tilde{v}_j(\omega) = \mathcal{O}(\omega)$ as $\omega \to 0$ since $\mathbf{v}(s)$ is the time derivative of the position $\mathbf{X}(s)$. Plugging (2.5) in the potential (2.4) and using the saddle-point approximation at large $\omega r$ then yields

$$A_i(\mathbf{x}, t) \sim \frac{q}{2\pi} \sqrt{\frac{\pi}{2r}} \int_{-\infty}^{+\infty} \mathrm{d}\omega \frac{\tilde{v}_i(\omega)}{\sqrt{\omega}}\, e^{i\omega u + i\pi/4}. \tag{2.6}$$

As for the scalar potential given by (2.1), it still satisfies (2.3) at leading order. Note the stark contrast between the Coulombic $\mathcal{O}(\ln r)$ fall-off of the latter, and the radiative $\mathcal{O}(r^{-1/2})$ fall-off in (2.6). The difference may be seen as a remnant of the saddle-point approximation, which only holds at $\omega \neq 0$ and was therefore not valid in the Coulombic case.

Equations (2.3) and (2.6) were written in inertial coordinates and in Lorenz gauge, but moving to Bondi coordinates is a simple matter: using as usual $x + iy = r\, e^{i\phi}$ and $t = u + r$, the Minkowski spacetime metric becomes

$$\mathrm{d}s^2 = -\mathrm{d}u^2 - 2\mathrm{d}u\, \mathrm{d}r + r^2 \mathrm{d}\phi^2 \qquad \text{(pure Minkowski)} \tag{2.7}$$

and Bondi components of the electromagnetic potential are related to its inertial components by

$$A_u = A_0 = \mathcal{O}(\ln r), \qquad A_\phi = xA_y - yA_x = \mathcal{O}(r^{1/2}), \qquad A_r = A_0 + \frac{1}{r}A_i x^i = \mathcal{O}(\ln r). \tag{2.8}$$

This is already close to the fall-offs announced in (1.1); the only difference is that Lorenz gauge and radial gauge differ. To move from the former to the latter, perform a gauge transformation $A \to A + \mathrm{d}\varepsilon$ that sets $A_r + \partial_r \varepsilon = 0$. This fixes $\varepsilon$ up to a $(u, \phi)$-dependent integration function that we set to zero so as to leave the components $A_u$ and $A_\phi$ unaffected. Note that $\varepsilon$ is guaranteed to only depend on $r$ at leading order, since the leading part of $A_r$ is the Coulombic field (2.1); as for subleading terms, one can actually check that the radiative scalar potential in Lorenz gauge satisfies $A_0 \sim -\frac{q}{2\pi} \ln(r) - \frac{1}{r}A_i x^i + \mathcal{O}(r^{-3/2})$, so that $A_r$ in (2.8) is independent of $(u, \phi)$ at least up to order $r^{-3/2}$. This guarantees that the gauge transformation moving from Lorenz gauge to radial gauge can indeed be chosen so as to leave $A_u$ and $A_\phi$ untouched, at least at leading order.

## 2.2 Fall-offs in Bondi gauge

Having determined the electromagnetic fall-offs that will be used throughout the paper, we now investigate them in detail and solve Maxwell's equations in Bondi coordinates, near null infinity, and in radial gauge $A_r = 0$. We stress that essentially identical results will hold even in the presence of gravitation (see section 3.2), so this is a key prerequisite for all that follows.

**Fall-offs.** We start with the detailed form of the fall-off conditions (1.1) for the Maxwell field:

$$A_u = \underbrace{E \ln r}_{\text{Coulombic}} + G + \sum_{m \in \mathbb{N}/2} \sum_{n=0}^{\lceil m \rceil} A_u^{m,n} \frac{(\ln r)^n}{r^m}, \tag{2.9a}$$

$$A_\phi = A_\phi^\ell \ln r + A_\phi^0 + \sum_{m \in \mathbb{N}/2} \sum_{n=0}^{\lceil m \rceil} A_\phi^{m,n} \frac{(\ln r)^n}{r^m} + \underbrace{C \sqrt{r}}_{\text{radiative}}. \tag{2.9b}$$

Here the sum over $m$ includes both integer and half-integer powers of $r$, $\lceil \cdot \rceil$ denotes the ceiling function,[2] and each component of the form $A_\times^\times$ is *a priori* an arbitrary function of $(u, \phi)$. Note the leading Coulombic data (electric charge aspect) $E(u, \phi)$ in (2.9a): this mimics the Coulomb solution (2.3) and the fall-off in (2.8), but now includes an angular dependence to account for the possibility of Lorentz boosts (and ultimately superrotations, as we shall see). Also note how $A_\phi$ in (2.9b) has an overleading term $C\sqrt{r}$ with respect to $A_u$, containing a field suggestively called $C(u, \phi)$ in analogy with the Bondi shear of 4D gravity. Indeed, $C$ will turn out to incorporate radiative degrees of freedom as in (2.8), and its time derivative (1.2) will be the electromagnetic news.

A remark is in order regarding the field $G(u, \phi)$ in (2.9a). Namely, the argument in $\ln r$ must be dimensionless, so an implicit radial constant $r_0$ was set to unity; the first two terms in (2.9a) should thus be understood as $E \ln(r/r_0) + G$. One could be tempted to absorb $G$ in the definition of $r_0$ by rescaling the radial coordinate, but we will keep $G(u, \phi)$ as a separate, independent field because it will ultimately appear in asymptotic charges.

**Equations of motion.** Starting with the fall-offs (2.9), one can solve the vacuum Maxwell equations $\nabla^\mu F_{\mu\nu} = 0$, which are always valid away from sources. The radial Maxwell equation $\nabla^\mu F_{\mu r} = 0$ then determines the scalar potential $A_u^{m>0,n}$ in terms of the vector potential $A_\phi$. More precisely, solving recursively in $1/r$, one gets the leading-order behaviour

$$\nabla^\mu F_{\mu r} = \mathcal{O}(r^{-3}) \quad \Rightarrow \quad \begin{cases} A_u^{1/2,1} = 0, \\ A_u^{1/2,0} = 2C', \end{cases} \tag{2.10}$$

while subleading components are relegated to appendix A.1. As for the angular Maxwell equation $\nabla^\mu F_{\mu\phi} = 0$, it determines the time evolution of the coefficients of $A_\phi$. Again solving recursively in $1/r$, one finds

$$\nabla^\mu F_{\mu\phi} = \mathcal{O}(r^{-2}) \quad \Rightarrow \quad \begin{cases} \dot{A}_\phi^\ell = E', \\ \dot{A}_\phi^0 = E' + G', \\ \dot{A}_\phi^{1/2,1} = 0, \\ \dot{A}_\phi^{1/2,0} = \frac{3}{2}\left(\frac{1}{4}C + C''\right). \end{cases} \tag{2.11}$$

---

[2]Note that nothing requires for now that the sums over $n$ in (2.9) stop at $n = \lceil m \rceil$. One could *a priori* consider expansions more general than (2.9). However, Maxwell's equations turn out to impose $A_{u,\phi}^{m,n>\lceil m \rceil} = 0$. We chose to include this piece of information as a defining property in (2.9).

These equations, along with (2.10), show that the whole dynamics is fixed by $C(u, \phi)$ along with the Coulombic data $E(u, \phi)$ and the function $G(u, \phi)$. Finally turning to the remaining Maxwell equation, one obtains

$$\lim_{r \to \infty} (r \nabla^\mu F_{\mu u}) = 0 \quad \Rightarrow \quad \dot{E} = 0. \tag{2.12}$$

Thus the Coulombic data is forced by dynamics to be time-independent: $E = E(\phi)$. This is in fact the only contribution in the temporal Maxwell equation. Indeed, since $\nabla^\mu F_{\mu r} = 0 = \nabla^\mu F_{\mu \phi}$ has already been solved, one can deduce from the identity $\nabla^\mu \nabla^\nu F_{\mu\nu} = 0$ that $\partial_r (r \nabla^\mu F_{\mu u}) = 0$, which means there is a single term in the radial expansion. Note that the solutions (2.10), (2.11) and (2.12) are all linear in the Maxwell fields, while the corresponding solutions in Einstein–Maxwell theory are non-linear due to gravitational backreaction (see appendix A).

**Gauge-invariant fields.** With the fall-offs (2.9), the components of the Faraday tensor are

$$B := F_{r\phi} = \frac{C}{2\sqrt{r}} + \frac{A_\phi^\ell}{r} + \mathcal{O}(r^{-3/2}), \tag{2.13a}$$

$$E_r := F_{ur} = -\frac{E}{r} + \frac{C'}{r^{3/2}} + \mathcal{O}(r^{-2}), \tag{2.13b}$$

$$E_\phi := F_{u\phi} = N\sqrt{r} + E' + \mathcal{O}(r^{-1/2}), \tag{2.13c}$$

where $N$ is the news (1.2). We stress again that the radiative and Coulombic data in the radial electric field (2.13b) do not appear at the same order, differently from what happens in 4D. The fact that the boundary conditions (2.9) include both Coulombic and radiative data can then also be seen from the Newman–Penrose formalism. Indeed, consider the triad of vectors

$$\ell := \partial_r, \qquad n := -\partial_u + \frac{1}{2}\partial_r, \qquad m := \frac{1}{r}\partial_\phi \tag{2.14}$$

such that $g^{\mu\nu} = \ell^\mu n^\nu + \ell^\nu n^\mu + m^\mu m^\nu$ be the inverse of the Minkowski metric (2.7) in Bondi coordinates. The triad is thus null and normalized, and contracting its elements with the Faraday tensor (2.13) yields the Maxwellian Newman–Penrose scalars

$$\Phi_0 := F_{\mu\nu}\ell^\mu m^\nu = \frac{C}{2r^{3/2}} + \mathcal{O}(r^{-2}), \tag{2.15a}$$

$$\Phi_1 := F_{\mu\nu}n^\mu \ell^\nu = \frac{E}{r} + \mathcal{O}(r^{-3/2}), \tag{2.15b}$$

$$\Phi_2 := F_{\mu\nu}n^\mu m^\nu = -\frac{N}{\sqrt{r}} + \mathcal{O}(r^{-1}), \tag{2.15c}$$

where $\Phi_1$ and $\Phi_2$ respectively identify Coulombic and radiative data.

For future use in the Einstein equations, let us compute the (symmetrized) electromagnetic energy-momentum tensor

$$T_{\mu\nu} = 2F_{\mu\alpha}F_\nu{}^\alpha - \frac{1}{2}g_{\mu\nu}F^{\alpha\beta}F_{\alpha\beta}. \tag{2.16}$$

Its components $T_{uu}$ and $T_{u\phi}$, written in Bondi coordinates, respectively measure energy fluxes and angular-momentum fluxes. In the case at hand, $g_{\mu\nu}$ is the Minkowski metric (2.7) and one finds

$$T_{uu} = \frac{2N^2}{r} + \mathcal{O}(r^{-3/2}), \qquad T_{u\phi} = -\frac{2NE}{\sqrt{r}} + \frac{2}{r}\big(NC' - EE'\big) + \mathcal{O}(r^{-3/2}). \tag{2.17}$$

Note how the energy density $T_{uu}$ involves the square of the news; this will eventually yield a Bondi mass loss formula for the dynamical metric. Also note that the angular momentum density $T_{u\phi}$ depends not only on the radiative field $C$, but also on the Coulombic data $E$. The same property holds in 4D Maxwell theory, and is usually considered as a puzzle since one naïvely expects no Coulombic data to contribute to the radiation of angular momentum [89, 104, 105]. Another puzzle, albeit one that is specific to 3D, is that $T_{u\phi}$ integrated on the asymptotic circle appears to be divergent owing to the measure $r\mathrm{d}\phi$. A resolution of both puzzles will come from the asymptotic charge (5.12a), which is finite and has a purely radiative flux sourced by the news.

## 2.3  Electromagnetic memory

We close this section with a short discussion of the electromagnetic 'kick' memory effect [91], partly to illustrate how 3D and 4D differ. Consider a particle with mass $m$ and charge $q$ whose path in spacetime traces some worldline with proper 3-velocity $v$. The particle is subjected to some external electromagnetic field $F_{\mu\nu}$, whereupon its proper acceleration is given by the Lorentz force

$$\dot{v}^\mu = \frac{q}{m} F^\mu{}_\nu \, v^\nu. \tag{2.18}$$

To see how this leads to memory, write the proper velocity as $v = \alpha\partial_u + \beta\partial_r + (\gamma/r)\partial_\varphi$. Using the Faraday tensor (2.13), the equations of motion (2.18) become

$$\dot{\alpha} \sim -\frac{E}{r}\alpha, \qquad \dot{\beta} \sim -\frac{\dot{C}}{\sqrt{r}}\gamma, \qquad \dot{\gamma} \sim -\frac{\dot{C}}{\sqrt{r}}\alpha \tag{2.19}$$

at leading order in $1/r$. Thus, the angular acceleration $\dot{\gamma}$ is determined by the news, as in 4D [91]; but in contrast to 4D, the effect is suppressed by a factor $1/\sqrt{r}$. For a non-relativistic particle with $\alpha \sim 1$, a finite burst of radiation thus yields a net change of angular velocity given by

$$\Delta\gamma \sim -\frac{\Delta C}{\sqrt{r}}, \tag{2.20}$$

where the right-hand side involves the integral of electromagnetic news:

$$\Delta C(\phi) := C(u = +\infty, \phi) - C(u = -\infty, \phi) = \int \mathrm{d}u \, N(u, \phi). \tag{2.21}$$

The velocity change (2.20) is non-zero at finite $r$, but it decays as $r \to \infty$. As a consequence, the memory effect (2.20) is unrelated to the leading surface charges of Maxwell theory, which typically involve integrals of the Coulombic (as opposed to radiative) data $E(\phi)$ over the celestial circle: see (4.10) below. We will see nevertheless that subleading electromagnetic charges involve contributions of the form $C'/\sqrt{r}$ (see (4.11) below). One may view this as yet another manifestation of the mismatch between radiative and Coulombic fall-offs in 3D. By contrast, 4D electromagnetic memory involves a net change in angular velocity that is finite at infinity [91] and may be seen as a vacuum transition under leading asymptotic symmetries.

# 3 Setup for Einstein–Maxwell theory

In this section, we gather the ingredients needed to study asymptotic symmetries in Einstein–Maxwell theory. Starting from the Lagrangian, we compute the Noether current and the equations of motion, which we then solve with the fall-offs (2.9) in order to characterize the solution space. We conclude with a digression on simple zero-mode configurations.

## 3.1 Lagrangian and Noether currents

Consider the Lagrangian density for 3D Einstein–Maxwell theory:

$$\mathcal{L} = \frac{1}{2}\sqrt{-g}\left(R - F^{\mu\nu}F_{\mu\nu}\right). \tag{3.1}$$

The equations of motion obtained by varying the corresponding action with respect to $A^\nu$ and $g^{\mu\nu}$ are the standard Maxwell and Einstein equations[3]

$$\nabla^\mu F_{\mu\nu} = 0, \qquad E_{\mu\nu} := G_{\mu\nu} - T_{\mu\nu} = 0, \tag{3.2}$$

with $G_{\mu\nu}$ the Einstein tensor and $T_{\mu\nu}$ the energy-momentum tensor (2.16).

The pre-symplectic potential $\theta$ is identified as usual from the variation of the Lagrangian. More precisely, one has $\delta\mathcal{L} = (\text{EOM})\delta(A_\mu, g_{\mu\nu}) + \partial_\mu\theta^\mu$ with

$$\theta^\mu = \frac{1}{2}\sqrt{-g}\left(g^{\alpha\beta}\delta\Gamma^\mu_{\alpha\beta} - g^{\mu\alpha}\delta\Gamma^\beta_{\alpha\beta} - 4F^{\mu\nu}\delta A_\nu\right). \tag{3.3}$$

This is the starting point of the covariant phase space derivation of Noether charges associated with gauge and diffeomorphism symmetries [106]. Indeed, any infinitesimal diffeomorphism $\xi$ accompanied by a gauge transformation $\varepsilon$ transforms the metric and the electromagnetic potential according to $\delta_{\xi,\varepsilon}g_{\mu\nu} = \pounds_\xi g_{\mu\nu}$ and $\delta_{\xi,\varepsilon}A_\mu = \pounds_\xi A_\mu + \partial_\mu\varepsilon$. The corresponding transformation of the Lagrangian is $\delta_{\xi,\varepsilon}\mathcal{L} = \pounds_\xi\mathcal{L} = \partial_\mu(\xi^\mu\mathcal{L})$, and the ensuing off-shell Noether current is $J^\mu = \theta^\mu[\delta_{\xi,\varepsilon}] - \xi^\mu\mathcal{L}$, where $\theta$ is given by (3.3). In the case at hand, one finds

$$J^\mu = \sqrt{-g}\,\xi_\nu\left(E^{\mu\nu} + 2A^\nu\nabla_\alpha F^{\mu\alpha}\right) + \frac{1}{2}\sqrt{-g}\,\nabla_\nu\left(\nabla^\nu\xi^\mu - \nabla^\mu\xi^\nu - 4F^{\mu\nu}(\xi^\alpha A_\alpha + \varepsilon)\right). \tag{3.4}$$

On-shell, the first term vanishes by virtue of the equations of motion (3.2). The remaining surface term involves the integrand of the Noether charge, namely the 'Komar–Maxwell aspect'

$$K^{\mu\nu}_{\xi,\varepsilon} = \frac{1}{2}\sqrt{-g}\left(\nabla^\nu\xi^\mu - \nabla^\mu\xi^\nu - 4F^{\mu\nu}(\xi^\alpha A_\alpha + \varepsilon)\right). \tag{3.5}$$

The latter will be used below to compute asymptotic charges.

## 3.2 Fall-offs and radiative solution space

This section is the Einstein–Maxwell analogue of section 2.2. We specify gauge and fall-off conditions, then explain the construction of the corresponding solution space.

---

[3]The tensor $E_{\mu\nu}$ of the Einstein equation in (3.2) has nothing to do with the electric charge aspect $E$ in (2.9a).

**Fall-offs and gauge conditions.** In order to solve the field equations, one first has to pick gauge and fall-off conditions for the metric and the Maxwell field. For the latter, we choose as before radial gauge $A_r = 0$ and the fall-offs (2.9). For the metric, we work in Bondi coordinates and consider the standard line element [107] that generalizes the pure Minkowski form (2.7):

$$\mathrm{d}s^2 = Ve^{2\beta}\mathrm{d}u^2 - 2e^{2\beta}\mathrm{d}u\,\mathrm{d}r + r^2(\mathrm{d}\phi - U\mathrm{d}u)^2, \tag{3.6}$$

where $(\beta, U, V)$ *a priori* depend on all three coordinates $(u, r, \phi)$. The metric is thus written in Bondi gauge, with the three gauge conditions $g_{rr} = 0 = g_{r\phi}$ and $g_{\phi\phi} = r^2$.

**Bondi hierarchy of field equations.** Our goal is now to determine the perturbative large-$r$ behaviour of the functions $(\beta, U, V)$ thanks to the equations of motion (3.2). This is detailed in appendix A. The construction follows the so-called Bondi hierarchy [6], where one first solves hypersurface constraint equations (which are differential equations in $r$), before listing the true evolution equations in retarded time $u$. This resolution of the field equations can be summarized as follows.

First, the Einstein field equation $E_{rr} = 0$ determines the radial expansion of $\beta$ in terms of the components of $A_\phi$. The constraint $E_{r\phi} = 0$ then fixes the radial expansion of $U$ in terms of $A_\phi$ and $A_u$, up to a radial integration constant that is ultimately identified with the angular momentum aspect $L(u, \phi)$. The radial Maxwell equation $\nabla^\mu F_{\mu r} = 0$ then yields the coefficients $A_u^{m>0,n}$ in terms of $(E, A_\phi)$. Finally, the remaining hypersurface constraint $E_{ru} = 0$ determines the radial expansion of $V$ up to a radial integration constant, namely the mass aspect $M(u, \phi)$. The angular component $\nabla^\mu F_{\mu\phi} = 0$ of the Maxwell equation is then a true evolution equation that determines the time evolution of $A_\phi$, aside from that of the leading term $C$, while the equation $\nabla^\mu F_{\mu u} = 0$ sets $\dot{E} = 0$ as in the earlier Minkowski-space result (2.12). Finally, $E_{u\phi} = 0$ and $E_{uu} = 0$ determine the evolution of angular momentum and mass [see (3.10) below], whereupon $E_{\phi\phi} = 0$ is trivially satisfied.

A remark: in the solution for $\beta$ and $U$, we have set to zero the two radial integration constants $\beta_0(u, \phi)$ and $U_0(u, \phi)$ appearing at order $r^0$. These functions parametrize the induced boundary metric on $\mathscr{I}^+$ and can be included in the solution space of 3D vacuum gravity [97]. It would be interesting to investigate the effects of these boundary fields in the case of Einstein–Maxwell theory, but this goes beyond our scope here.

**Solution space.** At the end of the day, with the gauge and fall-off conditions (2.9)–(3.6), the non-vanishing components of the metric in Bondi gauge are

$$g_{uu} = 2E^2(\ln r) + 2M + \mathcal{O}(r^{-1/2}), \tag{3.7a}$$

$$g_{ur} = -1 + \frac{C^2}{2r} + \mathcal{O}(r^{-3/2}), \tag{3.7b}$$

$$g_{u\phi} = \frac{8}{3}\sqrt{r}\,CE + 2(\ln r)EA_\phi^\ell + L + \mathcal{O}(r^{-1/2}), \tag{3.7c}$$

$$g_{\phi\phi} = r^2. \tag{3.7d}$$

The components of the Faraday tensor are still given by (2.13), as in Minkowski space. The fact that electromagnetic aspects of the problem are so weakly affected by gravity may be viewed as a consequence of the absence of local gravitational degrees of freedom. Note, for comparison with reference [1], that the dictionary between our notations and those of [1] is obtained with the replacements

$$E \to -\lambda, \qquad A_\phi^\ell \to \alpha, \qquad C \to 0, \qquad M \to \frac{\theta}{2}, \qquad L \to \frac{1}{2}N_{[1]} - \lambda\alpha. \qquad (3.8)$$

We stress again that the electromagnetic radiative data $C$ vanishes identically in [1].

As in section 2.2, gauge-invariant fields can readily be deduced from the results (2.13) and (3.7). Thus the $(uu)$ and $(u\phi)$ components of the stress-energy tensor are given again by (2.17), and manifestly affect the metric at leading order in $1/r$ since the components (3.7) only reduce to the standard ones of pure gravity [13] when $E = C = 0$. As for the Newman–Penrose formalism, the triad (2.14) should now be replaced by the one appropriate for the metric in Bondi gauge (3.6):

$$\ell = \partial_r, \qquad n = -e^{-2\beta}\left(\partial_u + \frac{V}{2}\partial_r + U\partial_\phi\right), \qquad m = \frac{1}{r}\partial_\phi. \qquad (3.9)$$

The corresponding Newman–Penrose Maxwell scalars have once again the expansion (2.15), as in Minkowski space at leading order.

An important feature of the solution space concerns the nature of the free data on $\mathscr{I}^+$. Indeed, the functions $C(u, \phi)$ and $G(u, \phi)$ in (2.9) do not satisfy any evolution equation in $u$: their time evolution on $\mathscr{I}^+$ is unconstrained. As mentioned in the introduction, the presence of two free functions where one expects a single one (since 3D Maxwell theory has a single local degree of freedom) may seem puzzling. From the energy flux (2.17) however, it is clear that $C(u, \phi)$ is the electromagnetic shear; by contrast, $G$ is a residual gauge degree of freedom akin to those that can be unfrozen in 4D pure gravity [53, 54] or in vacuum 3D gravity with a free boundary metric [96, 97]. Aside from this free data on $\mathscr{I}^+$, the solution space contains an infinite amount of data satisfying evolution equations in $u$: the mass $M(u, \phi)$, the angular momentum $L(u, \phi)$, the charge aspect $E(\phi)$, and the components of $A_\phi$ other than $C$. In particular, the mass and angular momentum aspects, *i.e.* the terms of order $r^0$ in (3.7a) and (3.7c), satisfy the evolution equations

$$\dot{M} = -2N^2, \qquad \dot{L} = M' + EE' + \frac{1}{2}(CN' - 3C'N). \qquad (3.10)$$

The time derivative (1.2), naturally identified as the news, is indeed the quantity appearing in the flux of energy-momentum tensor $rT_{uu} = 2N^2 + \mathcal{O}(r^{-1/2})$ given by (2.17). It is illuminating that the evolution of the mass aspect in (3.10) takes a form similar to the 4D Bondi mass loss of pure gravity:[4] it is sourced by the electromagnetic news and exhibits that 3D Einstein–Maxwell theory with radiative fall-offs is analogous to a dimensional reduction of 4D pure gravity. At the difference with the 4D case however, one should note that the fluxes on the right-hand side of the evolution equations (3.10) do not involve any soft term linear in the news.

---

[4]The evolution of angular momentum does as well, with the addition of a Coulombic term, but the comparison with the 4D equations is perhaps less striking.

The analogy with 4D pure gravity is further strengthened by the computation of the on-shell symplectic potential (3.3). Its radial and temporal components are indeed given by

$$\theta^r = \frac{1}{4}\delta\Big[4(\ln r)E^2 + 2E^2 + 2M - CN\Big] + \underbrace{N\delta C + E\delta G}_{\text{non-exact}} + \mathcal{O}(r^{-1/2}), \tag{3.11a}$$

$$\theta^u = \frac{C\delta C}{r} + \mathcal{O}(r^{-3/2}), \tag{3.11b}$$

where $\theta^r$ contains a non-exact symplectic flux with two contributions: the first, $N\delta C$, is the 3D analogue of the term $N_{ab}\delta C^{ab}$ in 4D pure gravity; the second, $E\delta G$, is sourced by the extra free field $G(u,\phi)$ in the solution space. This is the origin of the spurious flux identified in [1], which will appear in the asymptotic charges. For the time being, we keep both sources of flux non-vanishing; we will set $G$ to zero only later, when studying surface charges.

## 3.3  Zero-mode solutions and angular momentum

Having characterized the solution space, it is illustrative to study 'zero-mode solutions', *i.e.* configurations in which all fields on $\mathscr{I}^+$ are constant parameters. In pure AdS$_3$ gravity with Brown–Henneaux boundary conditions [58], such zero-modes span a 2D parameter space indexed by mass and angular momentum, containing in particular black holes [108], conical deficits and excesses, and certain spacetimes with closed time-like curves (see *e.g.* the phase diagram in [65, fig. 8.5]). The flat limit of these vacuum solutions respectively results in flat-space cosmologies, conical deficits and conical excesses, while solutions with closed time-like curves are washed out [61].

The solution space built above contains purely gravitational data (*i.e.* data that is also present in the vacuum case), namely the mass $M$ and the angular momentum $L$. In addition, it includes an infinite amount of Maxwell data: the shear $C$, the spurious field $G$, the Coulombic electric charge aspect $E$, and all the components in the radial expansion of $A_\phi$. When searching for zero-mode solutions, it is natural to switch this all off except for the electric charge $E$. One thus seeks exact solutions labelled by three constant parameters: the mass $M$, the angular momentum $L$, and the electric charge $E$. We will indeed see in section 4.2 that these are the parameters conjugate to time translations, rotations, and global U(1) transformations respectively.

One property of such zero-mode configurations deserves special attention.[5] Namely, when all the Maxwell data is switched off except the charge $E$, the (appended) equation of motion (A.15) imposes $EL = 0$, implying that the solution cannot have both non-zero angular momentum $L$ and non-zero electric charge $E$. One might be tempted to conclude that a spinning charged solution is necessarily radiative, but this is not so, since a consistent solution space can be obtained even when $C = 0$ [1]. Instead, all (A.15) requires is that solutions having both $L \neq 0$ and $E \neq 0$ include other modes of $A_\phi$; to the best of our knowledge, there is no closed-form expression for such solutions.

The same conclusion can be reached using a flat limit of charged Bañados–Teitelboim–Zanelli (BTZ) solutions in AdS$_3$. Momentarily letting $\Lambda = -1/\ell^2$, an exact solution to the field equations

---

[5]What follows is a digression on angular momentum that has few implications for the rest of the paper. The hasty reader may thus go straight to section 4.

$\nabla^\mu F_{\mu\nu} = 0 = G_{\mu\nu} + \Lambda g_{\mu\nu} - T_{\mu\nu}$ is indeed given in coordinates $(t, r, \phi)$ by [109]

$$A_\mu \mathrm{d}x^\mu = \frac{Q}{\sqrt{2(1 - L^2/\ell^2)}} (\ln r)(\mathrm{d}t - L\,\mathrm{d}\phi), \tag{3.12a}$$

$$\mathrm{d}s^2 = \left(N_r \frac{r^2}{\rho^2} + \rho^2 N_\phi^2\right) \mathrm{d}t^2 - N_r^{-1}\mathrm{d}r^2 + 2\rho^2 N_\phi\,\mathrm{d}t\,\mathrm{d}\phi + \rho^2\mathrm{d}\phi^2, \tag{3.12b}$$

where the radial coordinates $r$ and $\rho$ are related by

$$\rho^2 = r^2 + \frac{L^2}{1 - L^2/\ell^2}\left(M + Q^2(\ln r)\right) \tag{3.13}$$

and the shift components are

$$N_r = -\frac{r^2}{\ell^2} + M + Q^2(\ln r), \qquad N_\phi = -\frac{L}{\rho^2(1 - L^2/\ell^2)}\left(M + Q^2(\ln r)\right). \tag{3.14}$$

The free parameters in (3.12) are $(M, L, Q)$, respectively fixing the mass, the angular momentum and the charge of the black hole. In the neutral limit $Q = 0$, the metric (3.12b) reduces to that of a BTZ black hole [108, 110], whose 'standard' metric is obtained when using $\rho$ as the radial coordinate. Taking instead the flat limit $\ell \to \infty$ yields a solution of (3.2) with

$$A_\mu \mathrm{d}x^\mu = \frac{Q}{\sqrt{2}}(\ln r)(\mathrm{d}t - L\,\mathrm{d}\phi), \tag{3.15a}$$

$$\mathrm{d}s^2 = N_r\,\mathrm{d}t^2 - N_r^{-1}\mathrm{d}r^2 - 2LN_r\,\mathrm{d}t\,\mathrm{d}\phi + (r^2 + L^2 N_r)\mathrm{d}\phi^2, \tag{3.15b}$$

and $N_r = M + Q^2(\ln r)$. In order to put this line element in Bondi gauge (3.6), we introduce the new coordinate $u = t + f(r) - L\phi$ with $\partial_r f(r) = N_r^{-1}$, whereupon

$$A_\mu \mathrm{d}x^\mu = \frac{Q}{\sqrt{2}}(\ln r)\left(\mathrm{d}u - N_r^{-1}\mathrm{d}r\right), \qquad \mathrm{d}s^2 = N_r\,\mathrm{d}u^2 - 2\mathrm{d}u\,\mathrm{d}r + r^2\mathrm{d}\phi^2. \tag{3.16}$$

Here the component $A_r$ can be trivially gauged away since it only depends on $r$. As a result, in Bondi gauge, the flat limit of a charged BTZ black hole has no angular momentum, and is parametrized only by the mass $M$ and the electric charge $Q$. The angular momentum $L$ has indeed been reabsorbed from (3.15) in the change of coordinates that puts the metric in Bondi gauge.

This result may seem surprising at first: had we set $Q = 0$ from the outset, the angular momentum would have been reabsorbed just as well, seemingly contradicting the existence of flat-space cosmologies with non-zero angular momentum. But in fact, the simplification agrees with the fact that all solutions of 3D gravity are locally isometric to a maximally symmetric spacetime, implying the existence of a diffeomorphism that indeed changes angular momentum. To see this, start from the flat-space zero-mode metric with mass $M$,

$$\mathrm{d}s^2 = M\mathrm{d}u^2 - 2\mathrm{d}u\,\mathrm{d}r + r^2\mathrm{d}\phi^2, \tag{3.17}$$

and change coordinates according to[6]

$$u \mapsto u + \frac{1}{M}\left(L\phi - r + \sqrt{r^2 - \frac{L^2}{M}}\right), \qquad r \mapsto \sqrt{r^2 - \frac{L^2}{M}}, \qquad \phi \mapsto \phi - \frac{1}{\sqrt{M}}\mathrm{arctanh}\left(\frac{r\sqrt{M}}{L}\right).$$

(3.18)

Using this in (3.17) yields the new metric

$$\mathrm{d}s^2 = M\mathrm{d}u^2 - 2\mathrm{d}u\,\mathrm{d}r + 2L\mathrm{d}u\,\mathrm{d}\phi + r^2\mathrm{d}\phi^2,$$

(3.19)

which is now a flat-space zero-mode with *both* mass $M$ *and* angular momentum $L$. Crucially though, the transformation (3.18) is a 'large' diffeomorphism that affects asymptotic charges. The steps leading from (3.12) to (3.16) similarly involved a large diffeomorphism to bring the flat limit of the charged BTZ black hole in Bondi gauge, thereby explaining why angular momentum is ultimately absent from (3.16).

# 4 Asymptotic symmetries and charges

The solution space of a theory is a covariant version of its phase space [111]. In the case at hand, we have just obtained the covariant phase space of 3D Einstein–Maxwell theory containing both radiative and Coulombic data. We now investigate its (asymptotic) symmetries and derive the associated charges. As we shall see, the presence of radiation entails non-integrable terms whose influence on the charge algebra will be studied at length in section 5.

## 4.1 Symmetry generators and transformation laws

We start with the derivation of the asymptotic Killing vectors $\xi^\mu = (\xi^u, \xi^r, \xi^\phi)$, seen as generators of diffeomorphisms that preserve the gauge and fall-off conditions used in writing the metric (3.6). In particular, any such diffeomorphism must preserve the Bondi gauge choices $g_{rr} = 0 = g_{r\phi}$ and $g_{\phi\phi} = r^2$. Computing the relevant Lie derivatives thus imposes the following conditions:

$$\pounds_\xi g_{rr} = 0 \quad \Rightarrow \quad \xi^u = f,$$

(4.1a)

$$\pounds_\xi g_{r\phi} = 0 \quad \Rightarrow \quad \xi^\phi = g - f'\int_r^\infty \frac{e^{2\beta}}{\tilde{r}^2}\,\mathrm{d}\tilde{r} = g + \frac{f'}{r} + \mathcal{O}(r^{-2}),$$

(4.1b)

$$\pounds_\xi g_{\phi\phi} = 0 \quad \Rightarrow \quad \xi^r = r\big(Uf' - (\xi^\phi)'\big) = -rg' + f'' + \mathcal{O}(r^{-1/2}),$$

(4.1c)

where $f = f(u,\phi)$ and $g = g(u,\phi)$ are free functions at this stage. An additional constraint is that $\xi^\mu$ needs to preserve the fall-offs (3.6). Namely, the condition $g_{ur} = -1 + \mathcal{O}(r^{-1})$ requires $\dot{f} = g'$, and preserving $g_{u\phi} = \mathcal{O}(\sqrt{r})$ imposes $\dot{g} = 0$. In short,

$$f = T + ug', \qquad\qquad \dot{T} = 0 = \dot{g},$$

(4.2)

---

[6]We do not discuss the range of the coordinates since we are only interested in producing a new solution.

where $T(\phi)$ and $g(\phi)$ are arbitrary functions on the celestial circle, respectively generating super-translations and superrotations [13]. The remaining fall-off $g_{uu} = \mathcal{O}(\ln r)$ is automatically preserved.

Note that asymptotic Killing vector fields need *not* preserve the gauge and fall-off conditions used in the Maxwell field (2.9), since any change due to a diffeomorphism can be reabsorbed by a suitable gauge transformation. In particular, one needs to ensure that the radial gauge $A_r = 0$ is preserved when acting on $A$ with the allowed symmetries $\delta_{\xi,\varepsilon} A_\mu = \pounds_\xi A_\mu + \partial_\mu \varepsilon$. Requiring $\pounds_\xi A_r + \partial_r \varepsilon = 0$ with $\xi$ given by (4.1)–(4.2) thus leads to the gauge parameter

$$\varepsilon = \alpha(u,\phi) + f' \int_r^\infty \frac{e^{2\beta} A_\phi}{\tilde{r}^2} \mathrm{d}\tilde{r} = \alpha(u,\phi) + \frac{2}{\sqrt{r}} f' C + \mathcal{O}(r^{-1}). \tag{4.3}$$

Here $\alpha(u,\phi)$ is a free function on $\mathscr{I}^+$, generating 'large gauge transformations' of the Maxwell field.[7] We stress that its time dependence is unconstrained (which will come back to haunt us when defining charges). The remaining fall-off conditions in (2.9) are automatically satisfied. Asymptotic symmetry generators are thus labelled by a $\mathfrak{bms}_3$ vector field $(T,g)$ and a function $\alpha$ on $\mathscr{I}^+$. We shall return to this structure shortly.

**Transformation laws.** The asymptotic symmetries obtained here were found off-shell, *i.e.* regardless of the equations of motion. In practice, their action on covariant phase space is obtained by computing Lie derivatives and gauge transformations of the solution space of section 3.2. The resulting transformation laws of the various fields of interest can then be organized as follows. First, the transformations of the mass and angular momentum aspects read

$$\delta_\xi M = f\dot{M} + \underbrace{gM' + 2g'M - g'''}_{\text{coadjoint } \mathfrak{bms}_3} \underbrace{- g'E^2}_{\text{Maxwell}}, \tag{4.4a}$$

$$\delta_\xi L = \underbrace{f\dot{L} + gL' + 2g'L + 2f'M - f'''}_{\supset \text{ coadjoint } \mathfrak{bms}_3} \underbrace{-2g'EA_\phi^\ell - \frac{1}{4}g''C^2 + \frac{1}{2}f'CN}_{\text{Maxwell}}, \tag{4.4b}$$

where one recognizes the coadjoint representation of the $\mathfrak{bms}_3$ algebra with central charges $c_1 = 0$, $c_2 \neq 0$ [60], supplemented by corrections due to the time dependence of $M, L$ and the presence of electromagnetic boundary data. Second, the transformations of the electromagnetic shear and news read

$$\delta_{\xi,\varepsilon} C = \left(f\partial_u + g\partial_\phi + \frac{1}{2}g'\right)C, \qquad \delta_{\xi,\varepsilon} N = \left(f\partial_u + g\partial_\phi + \frac{3}{2}g'\right)N, \tag{4.5}$$

exhibiting the fact that $C$ and $N$ are $\mathfrak{bms}_3$ primary fields with respective weights $1/2$ and $3/2$ under superrotations. Note the half-integer labels due to half-integer powers in the fall-offs (2.9), reminiscent of the half-integer superrotation weights that are ubiquitous in 4D gravity [82, 112]. Finally, the transformation laws of the remaining leading components of the Maxwell field are most

---

[7]This function $\alpha(u,\phi)$ has nothing to do with the $u$ component of the proper velocity in section 2.3.

instructive when imposing the equations of motion. The logarithmic terms then transform as

$$\delta_{\xi,\varepsilon}A_\phi^\ell = \left(f\partial_u + g\partial_\phi + g'\right)A_\phi^\ell + f'E \approx \left(fE + gA_\phi^\ell\right)', \tag{4.6a}$$

$$\delta_{\xi,\varepsilon}E = \left(f\partial_u + g\partial_\phi + g'\right)E \approx \left(gE\right)', \tag{4.6b}$$

so they are both $\mathfrak{bms}_3$ primary fields with unit weight under superrotations, and they involve no U(1) gauge parameter. By contrast, the $\mathcal{O}(1)$ terms transform as

$$\delta_{\xi,\varepsilon}A_\phi^0 = \left(f\partial_u + g\partial_\phi + g'\right)A_\phi^0 - g'A_\phi^\ell + f'G + \alpha' \approx \left(fG + gA_\phi^0\right)' + fE' - g'A_\phi^\ell + \alpha', \tag{4.7a}$$

$$\delta_{\xi,\varepsilon}G = \left(f\partial_u + g\partial_\phi + g'\right)G - g'E + \dot{\alpha}, \tag{4.7b}$$

so they also have unit weight under superrotations, but they now involve the large gauge transformation $\alpha$, including its time derivative in $\delta G$. We will see in section 5 that this time derivative plays an important role in the reduced phase space where $G$ is set to zero.

**Asymptotic symmetry algebra.** It is straightforward to compute the Lie bracket of asymptotic symmetries, seen as vector fields on phase space. (The Poisson brackets of the corresponding canonical charges are a whole other problem, treated separately in section 5.) Indeed, in terms of the 'modified bracket' of [13], designed to take into account field-dependent parameters, the asymptotic symmetry generators satisfy the commutation relations $\left[\delta_{\xi_1,\varepsilon_1},\delta_{\xi_2,\varepsilon_2}\right] = -\delta_{\xi_{12},\varepsilon_{12}}$ where

$$\xi_{12} := \left[\xi_1,\xi_2\right]_* := \left[\xi_1,\xi_2\right] - \delta_{\xi_1}\xi_2 + \delta_{\xi_2}\xi_1, \qquad \varepsilon_{12} := \pounds_{\xi_1}\varepsilon_2 - \delta_{\xi_1,\varepsilon_1}\varepsilon_2 - (1 \leftrightarrow 2). \tag{4.8}$$

In the case at hand, all symmetry generators are field-independent, so the $\delta$ pieces of (4.8) all vanish. The Lie bracket of asymptotic symmetry generators thus reads

$$f_{12} = f_1 g_2' + g_1 f_2' - \delta_{\xi_1}f_2 - (1 \leftrightarrow 2), \tag{4.9a}$$

$$g_{12} = g_1 g_2' - \delta_{\xi_1}g_2 - (1 \leftrightarrow 2), \tag{4.9b}$$

$$\alpha_{12} = f_1 \dot{\alpha}_2 + g_1 \alpha_2' - \delta_{\xi_1,\varepsilon_1}\alpha_2 - (1 \leftrightarrow 2), \tag{4.9c}$$

showing that the asymptotic symmetry algebra is a semi-direct sum $\mathfrak{bms}_3 \loplus C^\infty(\mathscr{I}^+)$, where $\mathfrak{bms}_3$ acts by Carrollian conformal transformations on functions on $\mathscr{I}^+$. This is the same result as in [1], which is thus unaffected by our choice of relaxed boundary conditions including $C$. However, the presence of $C$ does affect asymptotic charges, as we now show.

## 4.2 Asymptotic charges introduced

Let $\xi$ be an asymptotic Killing vector field and let $\varepsilon$ be an asymptotic U(1) gauge parameter, both as described above. They generate symmetries of the radiative phase space and should therefore define 'canonical generators', meaning functions on phase space—or *charges*—that produce the corresponding transformations in terms of the Poisson bracket. The functional differential of these charges can be computed thanks to the Iyer–Wald formula [113], starting from the symplectic

potential (3.3) and the Komar–Maxwell aspect (3.5). The upshot is the following variational one-form in field space:[8]

$$\begin{aligned}
\not{\delta}\mathcal{Q}_\xi &= \delta K^{ur}_\xi - K^{ur}_{\delta\xi} + \xi^u \theta^r - \xi^r \theta^u \\
&\overset{\circ}{=} f\big(\delta M + \underbrace{2N\delta C}_{\text{non-integrable}}\big) + g\delta\Big(L + \frac{1}{4}(C^2)' - E\big(A^\ell_\phi + 2A^0_\phi\big)\Big) - 2\big(\alpha + \underbrace{fG}_{\text{non-integrable}}\big)\delta E + \mathcal{O}(r^{-1/2}). \quad (4.10)
\end{aligned}$$

The same charge is found with the Barnich–Brandt formalism [114], so there is no ambiguity for now. An aside on notation: for compactness, we will always work with charge 'aspects' such as (4.10), as opposed to actual charges. The latter are celestial integrals of the former, so this convention allows us to omit the integral sign $\oint_{S^1}$ throughout. Since integrations by parts in $\partial_\phi$ will nevertheless be needed, the symbol $\overset{\circ}{=}$ stands for equalities where all boundary terms in $\partial_\phi$ have been discarded. The notation thus allows us to integrate by parts without having to carry the integral sign.

Several comments are in order regarding the charge variation (4.10). First, the asymptotic charge $\lim_{r\to\infty} \not{\delta}\mathcal{Q}_\xi$ is finite. This is already a non-trivial result since (i) the fall-offs (2.9) are weaker than those of [1] due to the radiative field $C$, and (ii) the integrated angular momentum defined from the energy-momentum tensor (2.17) is itself divergent (recall that a measure factor $r$ is required when computing the integral). Second, assuming for now that $\delta f = \delta g = \delta\alpha = 0$ (*i.e.* all asymptotic symmetry generators are field-independent), the charge (4.10) contains two non-integrable contributions, one of which ($fG\delta E$) is unrelated to electromagnetic radiation. The charge thus requires a more careful study, deferred to the next section, in order to understand (i) how to extract its integrable part, (ii) whether the integrable part is conserved or not, and (iii) how to compute the charge algebra.

Finally, a comment on the 3D electromagnetic memory effect. We saw in (2.20) that the change in angular velocity is sourced by the radiative field $C$, but it is clear from the charge (4.10) that this memory term is unrelated to charges of large U(1) gauge transformations, since the latter only involve the Coulombic data $E$. There is thus no relation between memory and leading charges. However, one can push the radial expansion of the charge variation one order further to find

$$\not{\delta}\mathcal{Q}_\varepsilon = -2\varepsilon\delta\big(\sqrt{-g}F^{ur}\big) = -2\alpha\delta E + \frac{2}{\sqrt{r}}\big(\alpha\delta C' - 2f'C\delta E\big) + \mathcal{O}(r^{-1}), \quad (4.11)$$

where the $r^{-1/2}$ term now involves a pairing $\alpha\delta C'$ that is reminiscent of the change of velocity (2.20). One can then invert the operator $\partial_\phi$ (say in the space of functions with vanishing mean) and interpret the memory effect (2.20) as a transition between vacua whose *subleading* U(1) charges differ. A similar link between memory and vacuum transitions exists in 4D, except that it crucially involves leading surface charges [91]. Thus, 3D electromagnetic memory is related to surface charges, but the link between them is less tight than in the 4D infrared triangle [17]. To the best of our knowledge, this is the first explicit (yet very simple) example of such a mismatch. It begs the question of the possible matching between asymptotic symmetries, memories and soft theorems in 3D (as opposed to 4D). We hope to come back to this issue in future work.

---

[8]For compactness, we denote the charge simply by $\mathcal{Q}_\xi$ although it depends on both $\xi$ and the U(1) parameter $\varepsilon$. The notation $\not{\delta}\mathcal{Q}_\xi$ represents a one-form in field space that may or may not be (and generally is not) exact.

# 5  Brackets and charge algebras

We now turn to a detailed analysis of the charges (4.10) and their brackets, *i.e.* the corresponding charge algebras. This involves several deep problems. The first is that charges are not integrable in the presence of the free data $C$ and $G$, so a prescription is required to choose their integrable part and compute the bracket. The second is that (4.10) contains two sources of non-integrability, and it is unclear if both should be called 'fluxes' since only the contribution of shear and news ($N\delta C$) is identified as radiation. The study of the charge therefore requires a proper treatment of the remaining 'spurious flux' ($G\delta E$).

To explore these issues, we will start by reviewing the commonly used Barnich–Troessaert (BT) bracket [14] and pointing out its (known) split ambiguities. We will then fix these, at least partly, by the Wald–Zoupas (WZ) prescription, before briefly discussing the so-called Noether bracket [98] and finally turning to the Koszul bracket [100].

Several possibilities and ambiguities will thus be covered, but the main conclusion is as follows: the most reasonable approach seems to be to set $G = 0$ and use the Koszul bracket [100]. A virtue of this prescription is to yield a charge algebra that does *not* depend on the split between integrable part and flux, while the condition $G = 0$ ensures one can define charges that are conserved in the absence of news. With these properties, both physically and mathematically well-motivated, the resulting algebra exhibits a field-dependent cocycle (and is therefore strictly speaking a Lie algebroid). In summary:

> Setting $G = 0$, consider the integrable charges $Q_\xi$ in (5.12a).
> These are Wald–Zoupas charges satisfying $\dot{Q}_\xi = 0$ in the absence of news.
> Then using the Koszul bracket (5.27), which is crucially split-independent,
> charges represent the symmetry algebra with the field-dependent cocycle (5.15).

Let us now explore the various possibilities in detail.

## 5.1  Barnich–Troessaert bracket

The first proposal for charge brackets in the presence of fluxes was given by Barnich and Troessaert in [14]. The computation of the eponymous bracket requires a split of the charge between an integrable part $\delta Q_\xi$ and a flux piece $\Xi_\xi[\delta]$, both of which are variational one-forms on field space (with the integrable piece being the one that is exact). Such splits will play a key role throughout this section. In the case at hand, the most obvious split of the charge variation (4.10) is

$$\slashed{\delta} \mathcal{Q}_\xi \overset{\circ}{=} \delta Q_\xi + \Xi_\xi[\delta] + \mathcal{O}(r^{-1/2}), \tag{5.1}$$

with an integrable piece and a flux respectively given by[9]

$$Q_\xi = fM + g\left(L + \frac{1}{4}(C^2)' - E(A_\phi^\ell + 2A_\phi^0)\right) - 2\alpha E \qquad \text{(integrable charge)}, \qquad (5.2a)$$

$$\Xi_\xi[\delta] = 2f(N\delta C - G\delta E) \qquad\qquad\qquad\qquad\qquad \text{(flux)}. \qquad (5.2b)$$

Several comments are in order about this split. First, it reproduces the choice made in [1] when $C = 0$. Its integrable part (5.2a) exhibits the familiar supertranslation charge aspect $fM$, the U(1) charge aspect $-2\alpha E$, and the superrotation charge aspect $gL$ supplemented by contributions due to electromagnetic boundary data. Second, note that we are being agnostic at this stage about the difference between the two non-integrable contributions $N\delta C$ and $G\delta E$, treating them as fluxes on equal footing. Finally, note for future reference that the very validity of the split relies on the assumption that the asymptotic symmetry parameters $(f, g, \alpha)$ are all field-independent; we will come back to this subtle point when setting $G = 0$ to define WZ charges in section 5.2.

Splits of charges between integrable pieces and fluxes form the starting point of the charge algebra. Indeed, the BT bracket of charges is defined as [14]

$$\{Q_{\xi_1}, Q_{\xi_2}\}_{\text{BT}} := \delta_{\xi_2} Q_{\xi_1} + \Xi_{\xi_2}[\delta_{\xi_1}] \qquad (5.3)$$

and satisfies the key property that the charge algebra reproduces the algebra of asymptotic symmetry generators, possibly up to a 2-cocycle $K$:

$$\{Q_{\xi_1}, Q_{\xi_2}\}_{\text{BT}} = Q_{[\xi_1, \xi_2]_*} + K_{\xi_1, \xi_2}. \qquad (5.4)$$

In the case at hand, one can use (5.2) along with the transformation laws (4.4)–(4.7) to separately evaluate the two contributions on the right-hand side of (5.3). Using again equalities up to integration by parts on the celestial angle, one finds

$$\delta_{\xi_2} Q_{\xi_1} \overset{\circ}{=} Q_{[\xi_1, \xi_2]_*} - (f_1 g_2''' - f_2 g_1''') - (f_1 g_2' - f_2 g_1')E^2 + 2(f_1\dot\alpha_2 - f_2\dot\alpha_1)E$$
$$- 2g_1 E(f_2 G)' - 2f_1 f_2 N^2 - 2g_1 f_2 C'N - g_1' f_2 CN, \qquad (5.5a)$$

$$\Xi_{\xi_2}[\delta_{\xi_1}] \overset{\circ}{=} 2f_2(N\delta_{\xi_1}C - G\delta_{\xi_1}E)$$
$$\overset{\circ}{=} 2g_1 E(f_2 G)' + 2f_1 f_2 N^2 + 2g_1 f_2 C'N + g_1' f_2 CN, \qquad (5.5b)$$

where the bracket of asymptotic symmetry generators in (5.5a) is given by (4.9). This confirms that the charge algebra satisfies the expected relation (5.4), with the centreless bracket (4.9) and a cocycle that turns out to be

$$K_{\xi_1, \xi_2} \overset{\circ}{=} -\underbrace{(T_1 g_2''' - T_2 g_1''')}_{\text{standard } \mathfrak{bms}_3} - \underbrace{(f_1 g_2' - f_2 g_1')E^2 + 2(f_1\dot\alpha_2 - f_2\dot\alpha_1)E}_{\text{field-dependent piece}}. \qquad (5.6)$$

Here one may recognize the standard central extension pairing superrotations and supertranslations in $\mathfrak{bms}_3$ [60], and the $E$-dependent cocycle is identical to that of [1]. This field-dependence is

---

[9]Technically, the charge (5.2a) is the integra*ted* version of an exact (*i.e.* integrable) one-form in field space, but the 'integrable' terminology is more standard so we stick to it.

reminiscent of that of the BMS$_4$ charge algebra [14]. A key difference, however, is that the cocycle of 4D gravity involves the gravitational shear, while here it is due to the Coulombic data $E$.

The presence of field-dependent central extensions is typically expected to affect physics—*e.g.* through representations (say coadjoint and their unitary quantization) and the ensuing holographic dual. Accordingly, it is essential to understand one's leeway in writing down a cocycle such as (5.6). How is it affected by the choice of split (5.1)? Are there choices that cancel the field dependence? The remainder of this entire section is devoted to such questions.

**Split ambiguities of charge algebras.**  To probe how the BT cocycle is affected by the split (5.1), one can redefine the latter by shifting the integrable piece as

$$\delta Q_\xi + \Xi_\xi[\delta] = \underbrace{\delta(Q_\xi + S_\xi)}_{\delta \widetilde{Q}_\xi} + \underbrace{\Xi_\xi[\delta] - \delta S_\xi}_{\widetilde{\Xi}_\xi[\delta]}. \tag{5.7}$$

Here $S_\xi$ is any 'shift' functional on solution space that is linear in the asymptotic symmetry parameters $(f, g, \alpha)$. Changing the split in this way furnishes a new BT bracket (5.3):

$$\{\widetilde{Q}_{\xi_1}, \widetilde{Q}_{\xi_2}\}_{\mathrm{BT}} = \delta_{\xi_2} \widetilde{Q}_{\xi_1} + \widetilde{\Xi}_{\xi_2}[\delta_{\xi_1}] = \widetilde{Q}_{[\xi_1, \xi_2]_*} + \underbrace{K_{\xi_1, \xi_2} - S_{\xi_1, \xi_2}}_{\text{shifted cocycle } \widetilde{K}_{\xi_1, \xi_2}} \tag{5.8}$$

where the shift of the cocycle is given by $S_{\xi_1, \xi_2} := \delta_{\xi_1} S_{\xi_2} - \delta_{\xi_2} S_{\xi_1} + S_{[\xi_1, \xi_2]_*}$. The latter can be deduced from the transformation laws (4.4) once the shift $S_\xi$ is given. For example, consider the following list of shifts:

$$S_\xi = fEG \qquad \Rightarrow \qquad S_{\xi_1, \xi_2} \overset{\circ}{=} -(f_1 \dot{\alpha}_2 - f_2 \dot{\alpha}_1)E + (f_1 g_2' - f_2 g_1')E^2, \tag{5.9a}$$

$$S_\xi = fM \qquad \Rightarrow \qquad S_{\xi_1, \xi_2} \overset{\circ}{=} (f_1 g_2''' - f_2 g_1''') + (f_1 g_2' - f_2 g_1')E^2, \tag{5.9b}$$

$$S_\xi = TM \qquad \Rightarrow \qquad S_{\xi_1, \xi_2} \overset{\circ}{=} (T_1 g_2''' - T_2 g_1''') + (T_1 g_2' - T_2 g_1')(E^2 + 2uN^2). \tag{5.9c}$$

Note that the last expression involves the pure supertranslation $T(\phi)$ defined by (4.2). Each of these changes the value of the cocycle (5.6).

The effect of cocycle shifts such as (5.9) can be dramatic, possibly even cancelling any field-dependence. To see this, take the initial split (5.2) and the cocycle (5.6) as starting points. Then shift the integrable charge (5.2a) by $S_\xi = f(M - 2EG)$ as in (5.7). Using (5.9a)–(5.9b), the resulting charge algebra takes the form (5.4) with a cocycle that differs from (5.6) and reduces, instead, to $\widetilde{K}_{\xi_1, \xi_2} \overset{\circ}{=} -2(T_1 g_2''' - T_2 g_1''')$. This is no longer field-dependent, and even coincides with the standard $\mathfrak{bms}_3$ central extension [60]. All the field-dependence of (5.6) has been compensated by a mere redefinition of the charges! However, one should keep in mind that the resulting integrable charge in (5.7) is no longer conserved in the absence of news, even in the case $G = 0$. In this sense, asking for field-independent cocycles may be overly restrictive. Let us therefore study more closely the conservation criteria that can be imposed on integrable charges, regardless of whether the ensuing cocycles are field-dependent or not.

## 5.2   Wald–Zoupas splits

Following the WZ prescription [29–31], it is natural to single out the integrable part of the charge in (5.1) by the requirement that it be conserved on suitable vacuum configurations. This was investigated in 4D Einstein–Maxwell theory in [89], where the radiative vacuum can be defined unambiguously in terms of gravitational and electromagnetic news. In the present case, however, one has to face the issue of the presence of the *two* fields $C$ and $G$. Only the former has a clear physical interpretation in terms of electromagnetic radiation and gives rise to news $N = \dot{C}$. The vacuum condition that should be used to construct WZ charges is thus unclear: should one leave $G$ arbitrary, or instead set $G = 0$? Here we explore both options in turn.

**Wald–Zoupas split with $G \neq 0$.**   If one insists on keeping $G$ arbitrary (and generally non-zero), an immediate issue arises: the U(1) symmetry generator (4.3) and the charge (4.10) contain a function $\alpha(u, \phi)$ whose dependence on $u$ is arbitrary, so that it cannot appear in WZ charges (which would prevent any possibility of conservation). A way out is to exclude the contribution of $\alpha$ from the definition of integrable charges, and define 'vacua' as being solutions with vanishing news. This is indeed a consistent choice of vacuum, since the news' transformation (4.5) is homogeneous. Thus, starting from (5.2a), one may consider the shifted charge $\widetilde{Q}_\xi = Q_\xi + 2E(gA^0_\phi + \alpha)$ such that the $\alpha$-dependent piece in (4.10) is, by definition, part of the flux $\Xi_\xi$ in (5.1). Then the derivative $\partial_u \widetilde{Q}_\xi$ is proportional to the news, so $\partial_u \widetilde{Q}_\xi = 0$ when $N = 0$. This is a consistent choice and it can be used in the BT bracket (5.3), leading in particular to a complicated field-dependent cocycle (which we do not display but can be easily computed). The price to pay is that *all* electromagnetic charges vanish, by construction. This is arguably unnatural in a theory that purports to include electrically charged configurations. Let us therefore explore an alternative route that constrains the time evolution of $\alpha$ by setting $G = 0$.

**Wald–Zoupas split with $G = 0$.**   Setting $G = 0$ is physically acceptable since it does not prevent the solution space from describing radiation or electric charge. As can be seen in (4.7b), a consistency requirement with $G = 0$ is to fix the time dependence of the U(1) gauge parameter to

$$\alpha(u, \phi) = ug'E + \alpha_0(\phi). \tag{5.10}$$

Such a constrained time dependence only leaves out one arbitrary function $\alpha_0(\phi)$, reducing the algebra $\mathfrak{bms}_3 \dotplus C^\infty(\mathscr{I}^+)$ mentioned below (4.9) to its time-independent subalgebra, $\mathfrak{bms}_3 \dotplus C^\infty(S^1)$. The relevant brackets, however, are subtle: they become field-dependent even without dealing with surface charges. Indeed, the modified bracket (4.8) now yields

$$\alpha_{0,12} = g_1 \alpha'_{0,2} + f_1 g'_2 E - (1 \leftrightarrow 2), \tag{5.11}$$

which agrees with [1, eq. (5.1)] and replaces (4.9c). The algebra $\mathfrak{bms}_3 \dotplus C^\infty(S^1)$ is thus endowed with an 'exotic' field-dependent bracket.

This complication extends to surface charges: since the relation (5.10) is field-dependent, it cannot be blindly inserted in (5.2a). Instead, one must go back to the variational expression (5.1)

and properly integrate the term $\alpha\delta E$. Doing so leads to a new split (5.1) with

$$Q_\xi = fM + g\left(L + \frac{1}{4}(C^2)' - E\left(A_\phi^\ell + 2A_\phi^0\right)\right) - \left(ug'E + 2\alpha_0\right)E \qquad \text{(integrable charge),} \quad (5.12\text{a})$$

$$\Xi_\xi[\delta] = 2fN\delta C \qquad\qquad\qquad\qquad\qquad\qquad\qquad\qquad\qquad \text{(flux),} \quad (5.12\text{b})$$

where the integrable part satisfies a WZ conservation property.[10] This is because its time derivative

$$\dot{Q}_\xi \overset{\circ}{=} -2N\delta_\xi C \overset{\circ}{=} -2fN^2 + g(CN' - C'N) =: F_\xi \qquad\qquad (5.13)$$

vanishes in the absence of news. Note that defining a bracket of fluxes[11] as $\left\{F_{\xi_1}, F_{\xi_2}\right\} := \delta_{\xi_2} F_{\xi_1}$ yields

$$\delta_{\xi_2} F_{\xi_1} \overset{\circ}{=} F_{[\xi_1,\xi_2]_*} - \partial_u\left(2f_1 f_2 N^2 + 2g_1 f_2 NC' + g_1' f_2 NC\right), \qquad (5.14)$$

which implies that integrated fluxes on $\mathscr{I}^+$ represent the symmetry algebra, provided both field configurations at $\mathscr{I}_+^+$ and $\mathscr{I}_-^+$ (the endpoints of future null infinity) are vacua with $N = 0$. Such flux brackets are useful for celestial holography and soft theorems [37, 115], but they will also provide a useful extra criterion to be satisfied (or violated) by choices of splits (5.1). We will therefore return to flux brackets shortly.

With the condition $G = 0$ and the WZ split (5.12), the BT bracket represents the asymptotic symmetry algebra in the sense of (5.4), now with the aforementioned field-dependent bracket (5.11) and a new field-dependent cocycle

$$\boxed{K_{\xi_1,\xi_2} = -\underbrace{(T_1 g_2''' - T_2 g_1''')}_{\text{standard } \mathfrak{bms}_3} + \underbrace{(T_1 g_2' - T_2 g_1')E^2}_{\text{field-dependent piece}}.} \qquad (5.15)$$

This coincides with the earlier result (5.6) upon imposing $\dot{\alpha} = g'E$. The same cocycle will occur with the Koszul bracket in section 5.4, so we highlight it here to stress that it is a genuine split-independent feature of 3D Einstein–Maxwell theory.

Again, the actual form of the cocycle is not robust under changes of the split (5.1). For WZ charges, any shift (5.7) by a quantity which is conserved in the absence of news is allowed, since it leaves conservation unaffected. Consider for example the shift (5.9c) by $TM$, which is time-independent when $N = 0$. Then the WZ split (5.12) changes into

$$Q_\xi = (f + T)M + g\left(L + \tfrac{1}{4}(C^2)' - E\left(A_\phi^\ell + 2A_\phi^0\right)\right) - \left(ug'E + 2\alpha_0\right)E \qquad \text{(integrable),} \quad (5.16\text{a})$$

$$\Xi_\xi[\delta] = 2fN\delta C - T\delta M \qquad\qquad\qquad\qquad\qquad\qquad\qquad\qquad \text{(flux).} \quad (5.16\text{b})$$

---

[10] One can also view the integrable part as arising from the WZ formula $\delta Q_\xi + \Xi_\xi[\delta] + \xi \lrcorner \vartheta$, where $\vartheta$ is a WZ potential [29–31]. Choosing $\vartheta = (N\delta C)\partial_r$ yields $\xi \lrcorner \vartheta\big|_{S_\infty^1} = 2(\xi^r \vartheta^u - \xi^u \vartheta^r) = -2\xi^u \vartheta^r = -2fN\delta C$ on the celestial circle, so the WZ potential cancels the flux piece (5.12b) and leaves out the WZ charge (5.12a).

[11] The same word 'flux' refers to two different objects: non-integrable pieces in charge variations as in (5.1), and time derivatives of integrable charges as in (5.13). In practice, there should be no confusion since the meaning of the word is always clear from context.

With this new split, the time derivative of integrable charges is

$$\dot{Q}_\xi \stackrel{\circ}{=} -2(f+T)N^2 + g(CN' - C'N) =: F_\xi, \tag{5.17}$$

and vanishes as expected if $N = 0$. The corresponding cocycle in (5.4) is now

$$K_{\xi_1,\xi_2} = -2\underbrace{(T_1 g_2''' - T_2 g_1''')}_{\text{standard } \mathfrak{bms}_3} - \underbrace{2uN^2(T_1 g_2' - T_2 g_1')}_{\text{field-dependent piece}}, \tag{5.18}$$

which is remarkable: the cocycle is still field-dependent, but it becomes field-*independent* in the absence of news. This is nearly all one could hope for, but there is a catch: in contrast to the earlier fluxes (5.13), the new fluxes (5.17) fail to represent the symmetry algebra in the sense of (5.14). We conclude, at least in the present case, that *the following three requirements cannot be met simultaneously*:

(i) having integrable charges that are conserved in the absence of news;

(ii) having cocycles that are field-independent in the absence of news;

(iii) having fluxes that represent the symmetry algebra in the absence of news, as in (5.14).

**Changing slicings.** In the derivation of asymptotic symmetries in section 4, the parameters $(f, g, \alpha)$ are functions of $(u, \phi)$ and appear as radial integration constants. They can therefore be redefined in an arbitrary manner, possibly even a *field-dependent* one. This freedom is known as a 'change of slicing', and was studied at length in [96, 97, 116–118]. It can be used *e.g.* to reabsorb 'spurious fluxes' or 'fake news' appearing because of overly loose boundary conditions. For example, in [97] the authors studied vacuum 3D gravity with a free boundary metric on $\mathscr{I}^+$, where Iyer–Wald charges generally involve many non-integrable contributions despite the absence of local degrees of freedom; field-dependent redefinitions of the symmetry generators (*i.e.* changes of slicing) can then be used to cancel the spurious non-integrability.

How does this freedom affect the present discussion? In the charge variation (4.10), the two non-integrable contributions $fN\delta C$ and $fG\delta E$ stand on different footings: the former cannot be reabsorbed by a change of slicing, but the latter can, namely by redefining the U(1) parameter as

$$\alpha = \tilde{\alpha} - fG. \tag{5.19}$$

One may view this as an indication that the news represents physical flux, while $G$ is a spurious flux that can be removed. Assuming then that the new parameter $\tilde{\alpha}$ is field-independent ($\delta\tilde{\alpha} = 0$), the charge (4.10) splits into

$$Q_\xi = fM + g\left(L + \frac{1}{4}(C^2)' - E\big(A_\phi^\ell + 2A_\phi^0\big)\right) - 2\tilde{\alpha}E \qquad \text{(integrable charge)}, \tag{5.20a}$$

$$\Xi_\xi[\delta] = 2fN\delta C \qquad\qquad\qquad\qquad\qquad \text{(flux)}. \tag{5.20b}$$

Note that $\tilde{\alpha}$ here is generally time-dependent. Despite this, it satisfies the field-dependent commutation relations (5.11) of the time-independent case (with $\alpha_0$ replaced by $\tilde{\alpha}$). As for the BT bracket (5.3), it satisfies again the algebra (5.4) with the same cocycle (5.15) as in the case $G = 0$.

The prescription (5.20) thus leads to integrable charges in the absence of news, with the same algebra as WZ charges in the time-independent phase space with $G = 0$. However, this does *not* solve the issue of the arbitrary time dependence in $\tilde{\alpha}$. Indeed, the time evolution of the integrable part (5.20a) now reads

$$\dot{Q}_\xi \stackrel{\circ}{=} F_\xi\big|_{(5.13)} + 2E\Big(g(E+G)' + g'E - \dot{\tilde{\alpha}}\Big),\tag{5.21}$$

whose right-hand side is non-zero even without news. One may imagine cancelling the problematic second term by imposing the time evolution constraint $\dot{\tilde{\alpha}} = g(E+G)' + g'E$, but this reintroduces a field-dependence in $\tilde{\alpha}$, hence new sources of non-integrability and spurious fluxes... In this sense, changes of slicing do not cure the ambiguities present in the initial bare charge variation (4.10). In particular, while they allow for the removal of the spurious flux, they do not provide an unambiguous prescription for the time evolution of charges.

## 5.3 Noether split and bracket

For completeness, and in order to illustrate the proposal of [98] on a concrete example, let us study the so-called Noether split and the associated bracket. For this, we consider again the general case where $G \neq 0$ and the time dependence of the gauge parameter $\alpha$ in (4.3) is completely arbitrary. The Noether split consists in fixing the ambiguity in (5.1) by choosing the integrable part to be the Noether charge. The latter actually coincides with the Komar–Maxwell charge (3.5) for Einstein–Maxwell theory. Thus, the split is chosen using the $r$-independent parts of the Noether charge and flux,

$$K_\xi^{ur} \stackrel{\circ}{=} -2fE^2(\ln r) + Q_\xi + \mathcal{O}(r^{-1/2}),\tag{5.22a}$$

$$-K_{\delta\xi}^{ur} + \xi^u\theta^r - \xi^r\theta^u \stackrel{\circ}{=} +2fE^2(\ln r) + \Xi_\xi[\delta] + \mathcal{O}(r^{-1/2}),\tag{5.22b}$$

which explicitly yields

$$Q_\xi = f\left(\frac{1}{2}CN - E(E+2G)\right) + g\left(L + \frac{5}{4}(C^2)' - E(A_\phi^\ell + 2A_\phi^0)\right) - 2\alpha E \quad \text{(integrable)},\tag{5.23a}$$

$$\Xi_\xi[\delta] = f\left(\delta M + \frac{3}{2}N\delta C - \frac{1}{2}C\delta N + 2E\delta(E+G)\right) - g\delta(C^2)' \quad \text{(flux)}.\tag{5.23b}$$

With this split, the BT bracket (5.3) satisfies the charge algebra (5.4), with a homogeneous part given by (4.9) and a field-dependent cocycle[12]

$$K_{\xi_1,\xi_2} = 2(f_1g_2' - f_2g_1')(E^2 + CN).\tag{5.24}$$

This is remarkably simple, given the complicated shift that maps the initial split (5.2) on the Noether split (5.23). Note in particular that all Virasoro-type central extensions [involving third derivatives as in (5.6), (5.15) or (5.18)] have now disappeared.

---

[12]A clarification: what we call $K_\xi$ is the Noether charge (Komar) aspect (3.5), while $K_{\xi_1,\xi_2}$ is a cocycle in the charge algebra (5.4). The two objects are completely different, despite their similar notations.

The simplification of cocycles is in fact a built-in property of brackets of charges with the split (5.23), and it can be pushed further. Indeed, one can check that the cocycle (5.24) involves the on-shell Lagrangian density, in the sense that $K_{\xi_1,\xi_2} = \lim_{r\to\infty}(\xi_1^r \xi_2^u - \xi_1^u \xi_2^r)\mathcal{L}$. This is in fact a special case of a general result shown in [98]. Namely, when using the Noether charge $K_\xi$ and the 'Noetherian flux' defined as $\Xi_\xi[\delta] = -K_{\delta\xi} - \xi \lrcorner \theta$, the algebra (5.4) given by the BT bracket (5.3) produces a cocycle

$$K_{\xi_1,\xi_2} = \big(\xi_1 \lrcorner \xi_2 \lrcorner \mathcal{L}\big)\big|_{S^1_\infty} + a_{\xi_1,\xi_2} = \lim_{r\to\infty} \big(\xi_1^r \xi_2^u - \xi_1^u \xi_2^r\big)\mathcal{L} + a_{\xi_1,\xi_2}, \tag{5.25}$$

where $\mathcal{L}$ is the Lagrangian density and $a_{\xi_1,\xi_2}$ accounts for possible anomalies [30], which happen to be absent in the present setup. This leads to the so-called Noether bracket, defined in terms of the BT bracket (5.3) as

$$\big\{Q_{\xi_1}, Q_{\xi_2}\big\}_{\mathrm{N}} := \big\{Q_{\xi_1}, Q_{\xi_2}\big\}_{\mathrm{BT}} - \big(\xi_1 \lrcorner \xi_2 \lrcorner \mathcal{L}\big)\big|_{S^1_\infty} - a_{\xi_1,\xi_2}. \tag{5.26}$$

Using the charge algebra (5.4) with the cocycle (5.25), it is then immediate to show that the Noether bracket satisfies $\big\{Q_{\xi_1}, Q_{\xi_2}\big\}_{\mathrm{N}} = Q_{[\xi_1,\xi_2]_*}$. This represents the symmetry algebra with a *vanishing* cocycle! Of course, the simplification was bound to work, since (5.26) is just a BT bracket whose cocycle has been included in the very definition of the bracket. What crucially makes this possible is the explicit expression (5.25) of the cocycle, valid only for the Noether split.

Although the Noether split has the advantage of being unambiguously defined since it relies on the Noether charge, its drawback is to single out an integrable charge (5.23a) that is not conserved in the absence of news. An immediate way to see this is to note that no condition was imposed on the gauge parameter $\alpha(u, \phi)$. Alternatively, fixing $G = 0$ and the time dependence of $\alpha$ through (5.10) gives integrable Noether charges that slightly differ from (5.23a), but still satisfy $\dot{Q}_\xi \neq 0$ in the absence of news. The details are omitted here. Accordingly, we will not consider the Noether split any further, and now turn instead to one last choice of bracket.

## 5.4 Koszul bracket

Let us consider a proposal put forward and investigated by Barnich, Fiorucci, and Ruzziconi [100], to which we shall refer as the *Koszul bracket*. Given a split (5.1) between integrable part and flux, this bracket is defined as

$$\big\{Q_{\xi_1}, Q_{\xi_2}\big\}_{\mathrm{K}} := \big\{Q_{\xi_1}, Q_{\xi_2}\big\}_{\mathrm{BT}} - \int_\gamma \Big(\delta_{\xi_1} \Xi_{\xi_2}[\delta] - \delta_{\xi_2} \Xi_{\xi_1}[\delta] + \Xi_{[\xi_1,\xi_2]_*}[\delta]\Big). \tag{5.27}$$

Here the first term is the BT bracket (5.3), while the second term involves the integral, along a path $\gamma$, of a one-form in field space defined from the flux $\Xi_\xi[\delta]$. The path is such that its endpoint is the point in field space where the bracket (5.27) is meant to be evaluated. The starting point is arbitrary, and the specific choice of path is irrelevant thanks to the $\delta$-exactness of the integrand in (5.27). We will assume henceforth that some starting point has been chosen once and for all, independently of $\xi_1, \xi_2$.

We will neither derive nor investigate the properties of the Koszul bracket; suffice it to say that it is crucially independent of the choice of split (5.1). This is because the quantity $S_{\xi_1,\xi_2}$, produced

by the BT bracket under split changes (5.7), is exactly compensated by changes of the integral along $\gamma$ in (5.27) when shifting $\Xi_\xi[\delta] \to \Xi_\xi[\delta] - \delta S_\xi$. However, split-independence of (5.27) does not mean that the Koszul bracket only captures information about the genuine physical flux. Indeed, we will see that it is still sensitive to reductions of the solution space (*e.g.* by setting $G = 0$) and to changes of slicing.

We have already computed the BT bracket for the charges (5.2) and (5.12), respectively valid in the solution spaces with $G \neq 0$ and $G = 0$. The former leads to the cocycle (5.6), while the latter leads to (5.15), both of which are field-dependent. One can therefore ask how the second term in (5.27) affects the cocycle in these two cases. The answer turns out to be

$$\left(\delta_{\xi_1}\Xi_{\xi_2}[\delta] - \delta_{\xi_2}\Xi_{\xi_1}[\delta] + \Xi_{[\xi_1,\xi_2]_*}[\delta]\right)\Big|_{(5.2\mathrm{b})} = -\delta\Big((f_1 g_2' - f_2 g_1')E^2 - 2(f_1\dot\alpha_2 - f_2\dot\alpha_1)E\Big), \qquad (5.28\mathrm{a})$$

$$\left(\delta_{\xi_1}\Xi_{\xi_2}[\delta] - \delta_{\xi_2}\Xi_{\xi_1}[\delta] + \Xi_{[\xi_1,\xi_2]_*}[\delta]\right)\Big|_{(5.12\mathrm{b})} = 0, \qquad (5.28\mathrm{b})$$

showing that the change of cocycle due to (5.27) is entirely produced by the contribution of the spurious flux $-2fG\delta E$: it is non-zero in (5.28a) where $G \neq 0$, but vanishes in (5.28b) where $G = 0$. In particular, the correction (5.28a) exactly compensates the field-dependent part of the earlier cocycle (5.6), which thus reduces to the standard $\mathfrak{bms}_3$ central extension [60]. The Koszul bracket (5.27) entirely cancels the field-dependence of the cocycle when $G \neq 0$.[13] By contrast, the cocycle change (5.28b) *vanishes* for WZ charges in the solution space where $G = 0$, so the earlier field-dependent cocycle (5.15) is unaffected.[14] A third, mixed alternative is to consider the solution space $G \neq 0$ with the change of slicing (5.19) and flux (5.20b), whereupon (5.28b) holds again and yields again the earlier cocycle (5.15). In short, the cocycle in the Koszul bracket (5.27) only cares about whether $G$ is present or not in the flux $\Xi_\xi[\delta]$, and not about whether it was removed by a change of slicing or by the restriction $G = 0$.

This result exhibits the sensitivity of the Koszul bracket to the choice of slicing and to the 'size' of the solution space, despite its independence from the split (5.1). In particular, when $G \neq 0$, the Koszul bracket with fluxes (5.2b) yields a field-*independent* cocycle, while the change of slicing (5.19) reintroduces the field-dependent cocycle (5.15). The latter also appears for WZ charges with $G = 0$, whose algebra is unchanged by the Koszul bracket. This seems to indicate that the field-dependent cocycle (5.15) is a genuine feature of 3D Einstein–Maxwell theory, as was indeed suggested in [1]. It also begs, even more pressingly, the question of the interpretation of the field $G(u, \phi)$, which is ultimately responsible for most ambiguities of Einstein–Maxwell charge algebras.

---

[13]This holds up to a trivial central extension due to the field space integral in (5.27) [100].

[14]Note that setting $G = 0$ after having computed (5.28a) does not return the result (5.28b), since one cannot change the phase space *after* having computed the bracket.

# 6 Turning on $\Lambda \neq 0$

In order to deepen the analogy between our analysis and 4D pure gravity [53, 54], let us extend the setup to include a non-zero cosmological constant $\Lambda \neq 0$. The Bondi gauge is then well-adapted (at least in the vacuum case) for the flat limit $\Lambda \to 0$ [61, 97]. We therefore begin by solving the Einstein equations with cosmological constant $E_{\mu\nu} \coloneqq G_{\mu\nu} + \Lambda g_{\mu\nu} - T_{\mu\nu} = 0$, again with the fall-offs (2.9), then discuss asymptotic symmetries and their charges. Differences and similarities with the flat-space case $\Lambda = 0$ are pointed out throughout. We refer again to appendix A for some computational details.

## 6.1 Solution space and analogy with the 4D vacuum case

Here we solve the Einstein–Maxwell equations of motion with $\Lambda \neq 0$, then highlight the similarity between the resulting solution space and that of 4D Einstein gravity.

**Solution space.** Thanks to the Bondi gauge conditions $g_{rr} = 0 = g_{r\phi}$, the hypersurface equations $E_{rr} = 0$, $E_{r\phi} = 0$ and $\nabla^\mu F_{\mu r} = 0$ do not depend on $\Lambda$. As a result, they lead to the same solutions (A.3), (A.6) and (A.8) for $\beta$, $U$ and $A_u^{m>0,n}$ as in the flat case (recall that we have set $\beta_0 = 0 = U_0$). The first striking feature due to $\Lambda \neq 0$ comes from the angular Maxwell equation, which now reads

$$\nabla^\mu F_{\mu\phi} = -\frac{1}{4}\sqrt{r}\,\Lambda C - \frac{\Lambda}{4\sqrt{r}}\left((\ln r)A_\phi^{1/2,1} + \frac{3}{2}C^3 + A_\phi^{1/2,0} - 4A_\phi^{1/2,1}\right) + \mathcal{O}(r^{-1}) = 0. \qquad (6.1)$$

Note the leading term of this equation: it sets $C = 0$ on shell. The first subleading term then fixes $A_\phi^{1/2,1} = 0$, hence $A_\phi^{1/2,0} = 0$. Continuing with the expansion at subleading orders, one also finds $A_\phi^{3/2,0} = A_\phi^{3/2,1} = A_\phi^{3/2,2} = 0$, etc., which in turn implies [from (A.8)] that $A_u^{1/2,0} = A_u^{3/2,0} = A_u^{3/2,1} = 0$, etc. At the end of the day, the solution space with $\Lambda \neq 0$ contains no half-integer powers of $r$, and the fall-offs (2.9) simply require $m \in \mathbb{N}$ and $C = 0$.

The second interesting feature of $\Lambda \neq 0$ appears in the dynamical angular Maxwell equation. Indeed, expanding (6.1) further yields the equations[15]

$$\dot{A}_\phi^\ell = E' + \Lambda A_\phi^{1,1}, \qquad (6.2a)$$

$$\dot{A}_\phi^0 = E' + G' + \Lambda A_\phi^{1,0}, \qquad (6.2b)$$

$$3\dot{A}_\phi^{1,1} = 4\Lambda\left((A_\phi^\ell)^3 + A_\phi^{2,1}\right), \qquad (6.2c)$$

$$3\dot{A}_\phi^{1,0} = \frac{2\Lambda}{3}\left(6A_\phi^{2,0} - 2A_\phi^{2,1} + (A_\phi^\ell)^3\right) - 4MA_\phi^\ell + C^2E' + 2LE - 7E(C^2)' + 2(A_\phi^\ell)''. \qquad (6.2d)$$

When $\Lambda = 0$, these are evolution equations for the components of $A_\phi$ (apart from $C$, which is unconstrained), whose initial data is free and must be specified at some time $u_0$. There is an infinite tower of such equations, one for each $A_\phi^{m,n}$ in (2.9). But when $\Lambda \neq 0$, the meaning of these

---

[15]Note that we keep writing $C$ even though $C = 0$ when $\Lambda \neq 0$. This ensures we can still take the limit $\Lambda = 0$ and recover expressions that are valid in the flat case with $C \neq 0$. One should merely keep in mind that $\Lambda C = 0$.

equations changes radically due to the terms involving $\Lambda$ on the right-hand side of (6.2). These imply that, starting from $A_\phi^{1,0}$ and $A_\phi^{1,1}$, all the coefficients $A_\phi^{m,n}$ are algebraically determined by $A_\phi^0$ and $A_\phi^\ell$, which now become completely free data to be specified for all $u$. The temporal Maxwell equation then modifies the flat-space equation (A.16) to $\dot{E} = -\Lambda\big(A_\phi^\ell\big)'$, while $G$ remains completely free data as in the case $\Lambda = 0$. Finally, the evolution equations for mass and angular momentum take the form

$$\dot{M} = -2N^2 - \Lambda^2 A_\phi^\ell\big(2A_\phi^{1,0} + A_\phi^{1,1}\big) - \Lambda\Big(L' + 2A_\phi^\ell E' - E\big(A_\phi^\ell\big)'\Big), \tag{6.3a}$$

$$\dot{L} = M' + EE' + \frac{1}{2}\big(CN' - 3NC'\big) + \Lambda E\big(2A_\phi^{1,0} + A_\phi^{1,1}\big) - 2\Lambda A_\phi^\ell\big(A_\phi^\ell\big)', \tag{6.3b}$$

where again we have kept $C \neq 0$ for convenience in order to recover (3.10) in the flat limit. Note here that one can replace $A_\phi^{1,0}$ and $A_\phi^{1,1}$ by their value given by (6.2) in order to obtain the evolution of mass and angular momentum in terms of free data $(A_\phi^\ell, A_\phi^0, G)$.

Let us summarize. Starting with the fall-offs (2.9) where $C = 0$ and $m \in \mathbb{N}$, the solution space for $\Lambda \neq 0$ is completely determined by (i) three arbitrary functions of $(u, \phi)$ in $(A_\phi^\ell, A_\phi^0, G)$, along with (ii) the initial value at $u_0$ for the data $(M, L, E)$, subject to evolution equations (6.3) in $u$. There are thus two key differences with respect to $\Lambda = 0$. First, the free data on $\mathscr{I}^+$ is different: instead of $(C, G)|_{\Lambda=0}$, it is now given by $(A_\phi^\ell, A_\phi^0, G)|_{\Lambda \neq 0}$. Second, while $\Lambda = 0$ involves infinitely many evolution equations for the data in $A_\phi$, the case $\Lambda \neq 0$ only involves the three evolution equations for $(M, L, E)$.

**Analogy with 4D gravity.** The behaviour of 3D Einstein–Maxwell solution spaces at $\Lambda = 0$ and $\Lambda \neq 0$ is deeply reminiscent of the 4D vacuum case. A precise analogy can indeed be established thanks to the following three observations:

- First, the condition $\Lambda C = 0$ enforced by (6.1) is analogous to the condition obtained in 4D from the leading-order angular Einstein equations (see [53] or [54, eq. (2.17a)]). There, the constraint on the shear takes the form

$$\Lambda e^{2\beta_0} C_{ab} \propto (\partial_u - \partial_u \ln \sqrt{q}) q_{ab} + D_{\langle a} U_{b\rangle}^0, \tag{6.4}$$

  where the leading transverse metric $q_{ab}$ enters through its time dependence, while $\beta_0$ and $U_0^a$ are 4D versions of the integration constants that were discarded here.[16] The shear $C_{ab}$ thus depends on the induced metric on $\mathscr{I}^+$ when $\Lambda \neq 0$, and it must actually vanish if $U_0 = 0$ and if the time dependence of $q_{ab}$ is frozen, since then $\Lambda C_{ab} = 0$. The analogy with 3D Einstein–Maxwell is manifest and reinforces the interpretation of $C$ as the 3D electromagnetic analogue of the gravitational shear $C_{ab}$. Furthermore, it would be interesting to study the generalization of the condition $\Lambda C = 0$ with more relaxed 3D boundary conditions [96, 97], where one precisely has $U_0 \neq 0$ and a time-dependent metric on the celestial circle.

---

[16] The notation $D_{\langle a} U_{b\rangle}^0$ refers to the symmetric, trace-free part of the tensor $D_a U_b^0$.

- Next, note the analogy between square-root terms in the 3D Einstein–Maxwell system and logarithmic terms in 4D gravity. We saw here that $\Lambda \neq 0$ forces such square root terms to vanish. Similarly, subleading components of the angular Einstein equations in 4D imply that all logarithmic terms vanish [54], including both logarithmic terms in the angular metric, which can be introduced by hand through fall-offs [54, 119–121], and logarithmic terms that appear when solving for $U^a$ [107, 122].[17]

- The last part of the analogy involves the radial expansion of $A_\phi$ in 3D and that of the angular metric in 4D. Indeed, the latter reads

$$g_{ab} = r^2 q_{ab} + r C_{ab} + \frac{1}{4} q_{ab} C_{cd} C^{cd} + \sum_{n=1}^{\infty} \frac{g_{ab}^{(n)}}{r^n} \tag{6.5}$$

where we chose a regular, logarithm-free expansion. Then the leading angular Einstein equation constrains $C_{ab}$ as in (6.4), while the subleading terms in the angular equations are exactly analogous to the angular Maxwell equations (6.2) since they take the form [53]

$$\dot{g}_{ab}^{(n)} = \Lambda g_{ab}^{(n+1)} + (\dots). \tag{6.6}$$

When $\Lambda = 0$, these are evolution equations for the coefficients $g_{ab}^{(n)}$, whose value at some $u_0$ is required; there is an infinite tower of such equations. By contrast, when $\Lambda \neq 0$, the constraints (6.6) determine algebraically, and recursively, all the components $g_{ab}^{(n>1)}$ in terms of $g_{ab}^{(1)}$ and its time derivative. Then $g_{ab}^{(1)}$, rather than $C_{ab}$, is the free data that needs to be specified for all $u$, exactly like $A_\phi^\ell$ and $A_\phi^0$ become free data instead of $C$ in the 3D Einstein–Maxwell case.

To go beyond this simple analogy, one may attempt to use it to gain insights in 4D vacuum gravity. We saw in section 5 how the present 3D model allows one to study charges in the presence of radiation, as a first step towards holography with radiating sources. Another direction is to seek a $w_{1+\infty}$ symmetry of 3D Einstein–Maxwell theory. In 4D gravity, recent work has indeed shown the existence of such a structure [125–127]; it can be related to the tower of metric fields (6.5) and their evolution equations [128, 129]. Since 3D Einstein–Maxwell theory also admits such a hierarchy, with the components of $A_\phi$ playing the role of the subleading tensors in (6.5), it is natural to ask whether one can also exhibit an underlying higher-spin structure in that setup. We leave this investigation for future work.

## 6.2 Asymptotic symmetries and charges

We close this section with a brief discussion of asymptotic symmetries and charges in the case $\Lambda \neq 0$. Previous work along these lines can be found in [86] in Bondi gauge, and in [101, 102] in Fefferman–Graham coordinates with Hamiltonian methods. We will work in Bondi coordinates and

---

[17]This absence of logarithmic terms when $\Lambda \neq 0$ in 4D is a manifestation of the Starobinsky–Fefferman–Graham theorem [123, 124].

implement once and for all the condition of vanishing square root terms in the fall-offs, in particular setting $C = 0$. After going through the construction of the solution space, the metric is given by

$$g_{uu} = \Lambda r^2 + 2(\ln r)\left(E^2 - \Lambda\left(A_\phi^\ell\right)^2\right) + 2M - \Lambda\left(A_\phi^\ell\right)^2 + \mathcal{O}(r^{-1}), \tag{6.7a}$$

$$g_{ur} = -1 + \frac{\left(A_\phi^\ell\right)^2}{r^2} + \mathcal{O}(r^{-3}), \tag{6.7b}$$

$$g_{u\phi} = 2(\ln r)EA_\phi^\ell + L + \mathcal{O}(r^{-1}), \tag{6.7c}$$

$$g_{\phi\phi} = r^2, \tag{6.7d}$$

where the mass and angular momentum satisfy the evolution equations (6.3) with $C = 0$. It is manifest that these components reduce to those in (3.7) when $\Lambda = 0$.

The components of asymptotic Killing vectors are once again given by (4.1). Now, however, the functions $f$ and $g$ satisfy $\dot{f} = g'$ and $\dot{g} = -\Lambda f'$ instead of the flat-space relations (4.2). Their sum and difference $\sqrt{-\Lambda}f \pm g$ generate left- and right-moving conformal transformations. The ensuing transformations of metric and electromagnetic field components are as follows. First, mass and angular momentum transform as

$$\delta_\xi M = \left(f\partial_u + g\partial_\phi + 2g'\right)M - g''' - 2\Lambda f'L - g'E^2 + \Lambda A_\phi^\ell\left(f'E + g'A_\phi^\ell\right), \tag{6.8a}$$

$$\delta_\xi L = \left(f\partial_u + g\partial_\phi + 2g'\right)L - f''' + 2f'M - 2g'EA_\phi^\ell - \Lambda f'\left(A_\phi^\ell\right)^2, \tag{6.8b}$$

where one may recognize the coadjoint representation of the Virasoro algebra, written in terms of $M \propto T + \bar{T}$ and $L \propto T - \bar{T}$ instead of left and right stress tensor components $(T, \bar{T})$. This is accompanied by new contributions due to the Maxwell field. The latter has leading logarithmic components that transform as

$$\delta_{\xi,\varepsilon}A_\phi^\ell = \left(f\partial_u + g\partial_\phi + g'\right)A_\phi^\ell + f'E, \tag{6.9a}$$

$$\delta_{\xi,\varepsilon}E = \left(f\partial_u + g\partial_\phi + g'\right)E - \Lambda f'A_\phi^\ell, \tag{6.9b}$$

while its $\mathcal{O}(1)$ components transform in a way that involves the U(1) gauge parameter:

$$\delta_{\xi,\varepsilon}A_\phi^0 = \left(f\partial_u + g\partial_\phi + g'\right)A_\phi^0 - g'A_\phi^\ell + f'G + \alpha', \tag{6.10a}$$

$$\delta_{\xi,\varepsilon}G = \left(f\partial_u + g\partial_\phi + g'\right)G - g'E - \Lambda f'A_\phi^0 + \dot{\alpha}. \tag{6.10b}$$

All these equations reduce to the flat-space transformations (4.4)–(4.7) when $\Lambda = 0$.

One can also study the algebra of asymptotic symmetry transformations. Under the modified bracket (4.8), the algebra of the asymptotic Killing vectors is $\xi_{12} = \left[\xi_1, \xi_2\right]_*$, where one now has

$$f_{12} = f_1 g_2' + g_1 f_2' - \delta_{\xi_1}f_2 - (1 \leftrightarrow 2), \tag{6.11a}$$

$$g_{12} = g_1 g_2' - \Lambda f_1 f_2' - \delta_{\xi_1}g_2 - (1 \leftrightarrow 2), \tag{6.11b}$$

$$\alpha_{12} = f_1\dot{\alpha}_2 + g_1\alpha_2' - \delta_{\xi_1,\varepsilon_1}\alpha_2 - (1 \leftrightarrow 2) \tag{6.11c}$$

instead of the flat-space algebra (4.9). This is a semi-direct sum between two commuting copies of the Witt algebra and an algebra of functions on the cylinder or the plane. The flat limit ($\Lambda \to 0$) then behaves as an İnönü–Wigner contraction that maps the algebra (6.11) to the structure $\mathfrak{bms}_3 \dotplus C^\infty(\mathscr{I}^+)$ mentioned in section 4.1.

The charge aspect corresponding to asymptotic symmetry generators satisfies once again (5.1), albeit with a subleading correction $\mathcal{O}(r^{-1})$ instead of $\mathcal{O}(r^{-1/2})$. A simple choice of split is $e.g.$

$$Q_\xi = f\Big(M + \frac{\Lambda}{2}\big(A_\phi^\ell\big)^2\Big) + g\Big(L - E\big(A_\phi^\ell + 2A_\phi^0\big)\Big) - 2\alpha E \qquad \text{(integrable charge)}, \qquad (6.12\text{a})$$

$$\Xi_\xi[\delta] = 2f\big(\Lambda A_\phi^\ell \delta A_\phi^0 - G\delta E\big) \qquad\qquad\qquad \text{(flux)}, \qquad (6.12\text{b})$$

which reduces to the flat-space split (5.2) with $C = 0$ when $\Lambda = 0$. Using this split, the BT bracket (5.3) satisfies (5.4) with the homogeneous algebra (6.11) and the new field-dependent cocycle

$$K_{\xi_1,\xi_2} = -(f_1 g_2''' - f_2 g_1''') - (f_1 g_2' - f_2 g_1')E^2 + 2(f_1 \dot{\alpha}_2 - f_2 \dot{\alpha}_1)E$$
$$+ \Lambda\big(A_\phi^\ell\big)^2(f_1 g_2' - f_2 g_1') - 2\Lambda\big(EA_\phi^0 + A_\phi^\ell G\big)(f_1 f_2' - f_2 f_1'). \qquad (6.13)$$

The latter exhibits the standard Brown–Henneaux Virasoro extension [58], along with numerous new electromagnetic contributions. It reduces again to its flat-space counterpart (5.6) when $\Lambda = 0$.

In short, the entire structure reproduces the results of [1] in the flat limit $\Lambda = 0$. More detailed studies of the charges (6.12) and the WZ prescription are left for future work. This requires an understanding of radiation in the case $\Lambda \neq 0$, which we expect to share features and subtleties of the 4D vacuum case [52–57].

Let us conclude by returning to the Koszul bracket (5.27), now applied to surface charges (6.12) with $\Lambda \neq 0$. As in section 5.4, the Koszul bracket changes the cocycle (6.13) by a term of the form

$$\Big(\delta_{\xi_1}\Xi_{\xi_2}[\delta] - \delta_{\xi_2}\Xi_{\xi_1}[\delta] + \Xi_{[\xi_1,\xi_2]_*}[\delta]\Big)\Big|_{(6.12\text{b})} = -\delta K_{\xi_1,\xi_2}, \qquad (6.14)$$

where the variation $\delta$ only acts on the field-dependent terms in the cocycle (6.13). This is highly non-trivial: the Koszul bracket apparently cancels the entire field-dependence of the earlier extension (6.13), only leaving out its field-independent Brown–Henneaux piece! The same behaviour was observed in (5.28a) in the flat-space solution space with $G \neq 0$. In short, the presence of extra unconstrained boundary degrees of freedom leaves no trace in the Koszul bracket. The reason for this is unclear to us; we hope to return to it in future work, along with a more detailed analysis of the (A)dS$_3$ Einstein–Maxwell solution space.

## 7 Perspectives

This work was devoted to a detailed analysis of the asymptotic structure of 3D Einstein–Maxwell theory with radiative boundary conditions. It contributes in two specific ways to the already vast literature on asymptotic symmetries. First, it provides a model where gravitational asymptotic symmetries interplay in a non-trivial manner with radiative matter fields. Second, and most important, it does so in the simplest possible setup where all the subtleties related to radiative asymptotic

symmetries are present, thereby providing the most complete toy model one can hope for. We investigated this model along the following lines:

In section 2, we studied pure Maxwell theory in order to obtain the fall-offs (2.9) for Bondi-gauge radiative and Coulombic data, then solve the vacuum Maxwell equations. This revealed a key difference with previous work on 3D Einstein–Maxwell theory [1], namely that radiative data behaves as $\sqrt{r}$. Equipped with these fall-offs and with the line element (3.6) in Bondi gauge, we then solved the coupled equations of motion for Einstein–Maxwell theory in section 3. This enabled us to characterize the solution space and identify, in particular, the mass and angular momentum aspects with evolution equations (3.10). The latter confirm that 3D Einstein–Maxwell has radiative features akin to those of 4D pure gravity, albeit in a dimensionally-reduced setup. This was in fact expected, based on the prescient analysis of [59] where the first description of BMS$_3$ symmetries was done precisely through a dimensional reduction from the 4D theory. Here we supplemented this analysis with a complete control over the matter sector (either seen as arising from dimensional reduction, or put in by hand in 3D as we did).

Building on this setup, the core of our work in sections 4–5 consisted of an analysis of asymptotic symmetries and their representations by surface charges. For this, we first characterized the residual gauge transformations, spanning an algebra $\mathfrak{bms}_3 \oplus C^\infty(\mathscr{I}^+)$ that consists of diffeomorphisms and U(1) transformations. We then computed the charge variation (4.10) using the Iyer–Wald prescription. This revealed, as expected, that electromagnetic radiation described by the shear $C(u, \phi)$ gives rise to a non-integrable contribution, or flux. In addition, we found in agreement with [1] that the field $G(u, \phi)$ also sources non-integrability. The latter is somewhat spurious, so we treated $G$ as a gauge mode to be set to zero. Doing so reduced the asymptotic symmetry algebra because the U(1) gauge parameter acquires a fixed time dependence (5.10). It was then possible to construct WZ charges (5.12) which are conserved in the absence of news $N = \dot{C}$. As regards the algebra of such charges, we explored three alternatives for the bracket: the BT bracket, the Noether bracket, and the recently introduced Koszul bracket. The latter essentially extends the BT bracket in such a way that it does not depend on the split (5.1) between an integrable charge and a flux. Using this bracket, we showed that WZ charges with $G = 0$ represent the symmetry algebra with a *field-dependent* cocycle (5.17). This confirms the result of [1] while exhibiting both some of its robustness and some of its delicate dependence on the various apparently arbitrary choices involved in radiative surface charges.

In summary, our study has shown that 3D Einstein–Maxwell theory captures most of the subtleties related to radiative asymptotic symmetries. In addition to serving as a toy model of 4D gravity, it also reveals its own subtleties, the most striking of which is a field-dependent cocycle in the charge algebra. Whether such a field-dependent cocycle also occurs in 4D Einstein–Maxwell theory, say when using the Koszul bracket, is an open question which we keep for future work.

Having a complete control over the 3D setup, several interesting prospects are now in sight:

- **3D flat holography with radiation.** A natural follow-up of our construction of the Einstein–Maxwell solution space and its symmetries is to seek a candidate holographic description. The field-dependent cocycle (5.15) is expected to play a key role in that context, since it affects the structure of the symmetry group, which in turn affects coadjoint orbits

and unitary representations. In particular, it will be interesting to investigate how geometric actions for $BMS_3$ [69, 70, 73, 76–78] (or dual theories such as the one built in [130]) are modified by the coupling to electrodynamics. An alternative approach would be a 3D version of celestial holography, built by recasting the scattering amplitudes of 3D Maxwell theory in terms of conformal correlators on the celestial circle. Finally, it is natural to wonder how results on flat-space entanglement [131–133] generalize in the presence of matter fields.

- **3D Maxwell/scalar duality.** Since the 3D Maxwell field is dual to a scalar, a natural question is how this duality is implemented at the level of asymptotic charges. Indeed, such a puzzle also appears for 4D scalars whose soft theorem [134, 135] *a priori* has no asymptotic symmetry counterpart (scalars have no gauge symmetries). The puzzle is resolved by noting that a 4D scalar is dual to a 2-form gauge theory with non-trivial asymptotic symmetries [136–138]. In 3D Einstein–Maxwell theory, one faces the opposite situation, where the asymptotic structure of the Maxwell field is known but that of its dual is not. An approach that settles the issue in one stroke would be to study the Einstein–Maxwell-dilaton system. Relatedly, 3D Maxwell theory admits a gauge-invariant Chern–Simons mass term, presumably leading to rich asymptotic symmetries. This could also be a setup to analyse the peculiar properties of 3D quantum electrodynamics ($QED_3$), including the effects of its one-loop Chern–Simons mass term [139–142] on soft photons and asymptotic symmetries.

- **3D soft photons and memories.** Since 3D (Einstein–)Maxwell theory possesses a propagating massless photon, it admits a soft photon theorem [143] and, in principle, $(sub)^n$-leading soft photon theorems as well. Although such 3D soft photons have been mentioned in the literature [144], their detailed investigation has been superficial so far. What makes the issue especially attractive is that 3D Maxwell theory, being super-renormalizable, is badly infrared divergent as a result [145]. In that sense, the infrared problem in $QED_3$ is more violent than in 4D. This has partly been addressed using Faddeev–Kulish dressing [146], but a full picture of the infrared triangle [17] is still missing. We briefly mentioned this in sections 2.3 and 4.2 when pointing out that 3D electromagnetic memory fails to match with the leading U(1) charge. A natural continuation of our work is thus to ask how the soft photon theorem relates to memory and asymptotic symmetries.

- **4D Einstein–Maxwell.** Perhaps the most natural follow-up of the present work is to study the asymptotic structure of 4D Einstein–Maxwell theory. This has partly been explored in [89, 90], although the emphasis was on WZ charges in the first reference, and on dimensional reduction from 5D in the second one. The analysis can be vastly extended by studying the detailed structure of the solution space, including flux-balance laws encoding the soft graviton and soft photon theorems (or alternatively their memory counterpart). In fact, such a setup is ideal to investigate the interplay between soft photons and soft gravitons, which to our knowledge is an open question. In addition, one should compute the charge algebra using the Koszul bracket in order to see whether it gives rise to a field-dependent cocycle as in the present 3D model.

- **4D Einstein–Rosen waves.** As stressed throughout this work, the analogy between 3D Einstein–Maxwell and 4D pure gravity can be understood through the prism of dimensional reduction [59]. Interestingly, this dimensional reduction also reveals that the 3D theory actually describes a highly symmetric 4D system, namely cylindrical gravitational waves, or so-called Einstein–Rosen waves [87]. Since we have worked out the detailed asymptotic structure of 3D Einstein–Maxwell theory, it would now be interesting to reinterpret these results from the 4D vacuum point of view, where they could serve to investigate the radiative properties of Einstein–Rosen waves. This is potentially a setup where 4D holography in the presence of radiation is more approachable than in the case of arbitrary asymptotically flat spacetimes.

- **Hamiltonian approach.** A general question that stems from our work is whether ambiguities of surface charges can be fixed by avoiding the covariant formalism and using instead a Hamiltonian approach. The latter has indeed been recently successful at either re-deriving known 4D asymptotic symmetries [147–152] or unveiling new asymptotic structures altogether [153–156]. In the present case, one might attempt to recover the field-dependent central extension (5.15) from a canonical analysis, where integrability issues would be absent by design.

# Acknowledgements

We thank Glenn Barnich, Adrien Fiorucci and Romain Ruzziconi for key discussions on asymptotic symmetries with fluxes, as well as Mathieu Beauvillain, Steve Carlip, Daniel Grumiller, Ali Seraj, Shahin Sheikh-Jabbari and Simone Speziale for fruitful interactions. The work of J.B. was partially supported by the Swiss National Science Foundation through the NCCR SwissMAP. The work of S.M. is supported by the LABEX Lyon Institute of Origins (ANR-10-LABX-0066) within the Plan France 2030 of the French government operated by the National Research Agency (ANR). The work of B.O. is supported by the F.R.S.–FNRS and by the European Union's Horizon 2020 research and innovation programme under the Marie Skłodowska-Curie grant agreement No. 846244.

# A    Constructing the solution space

This appendix provides details on the construction of the Einstein–Maxwell solution space in sections 3.2 and 6.1. The construction follows the Bondi hierarchy [6], so that equations of motion are split into (i) hypersurfacce constraints, (ii) genuine evolution equations, and (iii) trivial equations that are automatically satisfied.

## A.1    Hypersurface equations

Within the Einstein–Maxwell equations of motion (3.2), consider those that are purely spatial and contain at least one radial index. We solve each of them in turn, using as initial inputs the Maxwell fall-offs (2.9), the stress tensor (2.16) and the Bondi metric ansatz (3.6).

**Einstein equation $(rr)$.** Begin by considering the field equation

$$E_{rr} = \frac{2}{r^2}\left(r\partial_r\beta - F_{r\phi}^2\right) = 0. \tag{A.1}$$

In radial gauge with $A_r = 0$, this determines $\beta$ in terms $A_\phi$ (and no other Maxwell component), with $A_\phi$ given by (2.9b). The equation is thus solved to order $E_{rr} = \mathcal{O}(r^{-5})$ by the expansion

$$\beta = \beta_0 + \frac{\beta_{1,0}}{r} + \frac{\beta_{3/2,0}}{r^{3/2}} + \frac{1}{r^2}\left(\beta_{2,0} + (\ln r)\beta_{2,1}\right) + \frac{1}{r^{5/2}}\left(\beta_{5/2,0} + (\ln r)\beta_{5/2,1}\right) + \mathcal{O}(r^{-3}), \tag{A.2}$$

where $\beta_0(u,\phi)$ is an integration constant that we henceforth set to zero, while the subleading terms are

$$\beta_{1,0} = -\frac{1}{4}C^2, \tag{A.3a}$$

$$\beta_{3/2,0} = -\frac{2}{3}CA_\phi^\ell, \tag{A.3b}$$

$$\beta_{2,1} = \frac{1}{4}CA_\phi^{1/2,1}, \tag{A.3c}$$

$$\beta_{2,0} = -\frac{1}{2}\left(A_\phi^\ell\right)^2 + \frac{1}{8}C\left(2A_\phi^{1/2,0} - 3A_\phi^{1/2,1}\right), \tag{A.3d}$$

$$\beta_{5/2,1} = \frac{2}{5}\left(A_\phi^\ell A_\phi^{1/2,1} + CA_\phi^{1,1}\right), \tag{A.3e}$$

$$\beta_{5/2,0} = \frac{2}{25}\left(A_\phi^\ell\left(5A_\phi^{1/2,0} - 8A_\phi^{1/2,1}\right) + C\left(5A_\phi^{1,0} - 3A_\phi^{1,1}\right)\right). \tag{A.3f}$$

Note that (A.3a) is reminiscent of the analogous link between shear and the function $\beta$ in 4D gravity in Bondi gauge.

**Einstein equation $(r\phi)$.** Still using the stress tensor (2.16) and the Bondi metric (3.6), consider the Einstein equation

$$E_{r\phi} = \frac{\beta'}{r} - \partial_r\beta' + \frac{r}{2}e^{-2\beta}\left((3 - 2r\partial_r\beta)\partial_r U + r\partial_r^2 U\right) - 2e^{-2\beta}F_{r\phi}(UF_{r\phi} + F_{ru}) = 0. \tag{A.4}$$

This is solved to order $E_{r\phi} = \mathcal{O}(r^{-3})$ by

$$U = U_0 + \frac{U_{3/2,0}}{r^{3/2}} + \frac{1}{r^2}\left(U_{2,0} + (\ln r)U_{2,1} + (\ln r)^2 U_{2,2}\right) + \frac{1}{r^{5/2}}\left(U_{5/2,0} + (\ln r)U_{5/2,1}\right) + \mathcal{O}(r^{-3}), \tag{A.5}$$

where $U_0(u, \phi)$ is undetermined while the subleading terms are

$$U_{3/2,0} = -\frac{8}{3}CE, \tag{A.6a}$$

$$U_{2,2} = \frac{1}{4}CA_u^{1/2,1}, \tag{A.6b}$$

$$U_{2,1} = -2A_\phi^\ell E + \frac{1}{2}C\left(A_u^{1/2,0} - \frac{3}{2}A_u^{1/2,1} - 2C'\right), \tag{A.6c}$$

$$U_{2,0} = -L, \tag{A.6d}$$

$$U_{5/2,1} = -\frac{8}{5}\left(EA_\phi^{1/2,1} + A_\phi^\ell A_u^{1/2,1} + CA_u^{1,1}\right), \tag{A.6e}$$

$$U_{5/2,0} = \text{(lengthy)}. \tag{A.6f}$$

Note that (A.4) is second-order in radial derivatives of $U$, so one gets two integration constants here. The first is $L(u, \phi)$ in (A.6d), eventually identified with the bare angular momentum aspect. The second is the aforementioned $U_0(u, \phi)$ in (A.5), which is analogous to $\beta_0$ in (A.2) since it specifies the induced boundary metric on $\mathscr{I}^+$; we always set $U_0 = 0$ from now on. Also note that we omit writing the component (A.6f) because its form is long without being particularly illuminating; it is included here merely for consistency, because the Einstein equation at order $E_{r\phi} = \mathcal{O}(r^{-3})$ does fix the component $U_{5/2,0}$.

**Maxwell equation $(r)$.** Now turn to the radial Maxwell equation $\nabla^\mu F_{\mu r} = 0$, which only depends on the metric through $\beta$ and $U$. Owing to the Bondi metric (3.6), its explicit form is

$$r^2 \nabla^\mu F_{\mu r} = F'_{\phi r} + re^{-2\beta}\left(\partial_r\left(r(UF_{r\phi} + F_{ru})\right) - 2r\partial_r\beta(UF_{r\phi} + F_{ru})\right) = 0. \tag{A.7}$$

This equation determines the subleading terms $A_u^{m>0,n}$ of the electrostatic potential in terms of $A_\phi$ and the Coulombic data $E$. More precisely, solving recursively in $1/r$, one gets the leading-order behaviour

$$\nabla^\mu F_{\mu r} = \mathcal{O}(r^{-3}) \quad \Rightarrow \quad \begin{cases} A_u^{1/2,1} = 0, \\ A_u^{1/2,0} = 2C' \end{cases} \tag{A.8}$$

which coincides with the Minkowskian Maxwell equations (2.10). As for the subleading components, they satisfy non-linear relations

$$\nabla^\mu F_{\mu r} = \mathcal{O}(r^{-4}) \quad \Rightarrow \quad \begin{cases} A_u^{1,1} = 0, \\ A_u^{1,0} = \left(A_\phi^\ell\right)' - \frac{5}{6}C^2 E, \\ A_u^{3/2,2} = 0, \\ A_u^{3/2,1} = -\frac{2}{9}\left(3CEA_\phi^\ell + \left(A_\phi^{1/2,1}\right)'\right), \\ A_u^{3/2,0} = \text{(lengthy)}. \end{cases} \tag{A.9}$$

Note, as in (A.6f), that we omit to write the long and obscure expression of $A_u^{3/2,0}$; it is merely mentioned here for consistency when solving Maxwell's equation $\nabla^\mu F_{\mu r} = 0$ at order $r^{-4}$.

The identifications (A.9) should be contrasted with their pure electromagnetic analogue: with the fall-offs (2.9), Maxwell's equations without sources would yield the *linear* relations

$$
\nabla^\mu F_{\mu r} = \mathcal{O}(r^{-4}) \quad \Rightarrow \quad
\begin{cases}
A_u^{1,1} = 0, \\
A_u^{1,0} = (A_\phi^\ell)', \\
A_u^{3/2,2} = 0, \\
A_u^{3/2,1} = -\dfrac{2}{9}(A_\phi^{1/2,1})', \\
A_u^{3/2,0} = -\dfrac{2}{9}(A_\phi^{1/2,0})' + \dfrac{4}{27}(A_\phi^{1/2,1})'.
\end{cases}
\tag{A.10}
$$

The non-linear contributions in (A.9) are due to gravitational backreaction on the Maxwell field.

**Einstein equation $(ru)$.** Let us finally turn to the last hypersurface equation:

$$
E_{ru} = \frac{1}{r^2}\left( VF_{r\phi}^2 - rV\partial_r\beta + \frac{r}{2}\partial_r V + rU(r\partial_r\beta - \beta)' + \frac{1}{2}\partial_r(r^2 U') - e^{2\beta}(\beta'^2 + \beta'') \right)
$$
$$
+ e^{-2\beta}(U^2 F_{r\phi}^2 - F_{ur}^2) + \frac{r}{4}e^{-2\beta}\left( (2r\partial_r\beta - 3)\partial_r U^2 - 2rU\partial_r^2 U - r(\partial_r U)^2 \right) = 0. \tag{A.11}
$$

The solution is given to order $E_{ru} = \mathcal{O}(r^{-3})$ by

$$
V = (\ln r)V_{0,1} + V_{0,0} + \frac{V_{1/2,0}}{r^{1/2}} + \mathcal{O}(r^{-1}), \tag{A.12}
$$

with

$$
V_{0,1} = 2E^2, \tag{A.13a}
$$
$$
V_{0,0} = 2M, \tag{A.13b}
$$
$$
V_{1/2,0} = \frac{8}{3}\big(2EC' - CE'\big). \tag{A.13c}
$$

The equation of motion (A.11) is first-order in radial derivatives of $V$, so it contains a single integration constant $M(u,\phi)$ in (A.13b), which is eventually identified with the bare mass aspect.

## A.2 Evolution equations

Now consider the dynamical part of the equations of motion (3.2), containing angular or time components, but no radial component. Again, we treat each such equation in turn.

**Maxwell equation $(\phi)$.** Start with the Maxwell equation $\nabla^\mu F_{\mu\phi} = 0$, which determines the time evolution of the coefficients of $A_\phi$. Solving recursively in $1/r$, one finds the leading-order evolution

equations

$$\nabla^\mu F_{\mu\phi} = \mathcal{O}(r^{-2}) \quad \Rightarrow \quad \begin{cases} \dot{A}^\ell_\phi = E', \\ \dot{A}^0_\phi = E' + G', \\ \dot{A}^{1/2,1}_\phi = -\dfrac{3}{4}CE^2, \\ \dot{A}^{1/2,0}_\phi = \dfrac{1}{4}\Big(7CE^2 - 3CM + 6C''\Big), \end{cases} \tag{A.14}$$

while at subleading order one has

$$\nabla^\mu F_{\mu\phi} = \mathcal{O}(r^{-5/2}) \quad \Rightarrow \quad \begin{cases} \dot{A}^{1,1}_\phi = 0, \\ \dot{A}^{1,0}_\phi = \dfrac{1}{3}\Big(C^2 E' - 4MA^\ell_\phi + 2E(L - 7CC') + 2(A^\ell_\phi)''\Big). \end{cases} \tag{A.15}$$

**Maxwell equation (u).** Turning to the remaining Maxwell equation, one gets

$$\lim_{r\to\infty} (r\nabla^\mu F_{\mu u}) = 0 \quad \Rightarrow \quad \dot{A}^\ell_u = 0. \tag{A.16}$$

This is in fact the only contribution in the temporal Maxwell equation. Indeed, using the fact that $\nabla^\mu F_{\mu r} = 0 = \nabla^\mu F_{\mu\phi}$ have already been solved, along with the form of the metric in Bondi gauge, one deduces from $\nabla^\mu \nabla^\nu F_{\mu\nu} = 0$ that $\partial_r(r\nabla^\mu F_{\mu u}) = 0$, which means that there is a single term in the radial expansion.

**Einstein equations (u$\phi$) and (uu).** Finally, consider the Einstein evolution equations $E_{u\phi}$ and $E_{uu}$. For the former, one finds

$$\lim_{r\to\infty} (rE_{u\phi}) = 0 \quad \Rightarrow \quad \dot{L} = M' + EE' + \frac{1}{2}\big(CN' - 3NC'\big), \tag{A.17}$$

which yields the evolution equation of angular momentum in (3.10). Similarly, one has

$$\lim_{r\to\infty} (rE_{uu}) = 0 \quad \Rightarrow \quad \dot{M} = -2N^2, \tag{A.18}$$

which is the Bondi mass loss equation in (3.10).

The point of the Bondi hierarchy is the following: it turns out that all subleading equations in $E_{u\phi}$ and $E_{uu}$ are automatically satisfied on-shell by virtue of the previous equations. We now show this using Bianchi identities.

## A.3 Trivial equations

Using the Bianchi identity for the Einstein tensor, one can write

$$0 = 2\sqrt{-g}\,\nabla_\mu G^\mu_\nu = 2\sqrt{-g}\,\nabla_\mu(E^\mu_\nu + T^\mu_\nu) = 2\partial_\mu(\sqrt{-g}\,E^\mu_\nu) + \sqrt{-g}\,E_{\rho\sigma}\partial_\nu g^{\rho\sigma} + 2\sqrt{-g}\,\nabla_\mu T^\mu_\nu. \tag{A.19}$$

Since all three Maxwell equations have been solved at this stage, the stress tensor is conserved, and the last term above drops. Since $E_{r\mu} = 0$ has already been solved as well, putting $\nu = r$ in (A.19)

and using $g^{uu} = 0 = g^{u\phi}$ yields $E_{\phi\phi}\partial_r g^{\phi\phi} = 0$. Thus the Bianchi identity automatically implies $E_{\phi\phi} = 0$ when $E_{r\mu} = 0$ and when Maxwell's equations are satisfied.

Finally, for $\nu = (\phi, u)$, the Bianchi identity (A.19) gives $\partial_\mu(\sqrt{-g}\, E_\phi^\mu) = 0 = \partial_\mu(\sqrt{-g}\, E_u^\mu)$. Since by now $E_{r\phi} = 0 = E_{\phi\phi}$, the first of these identities reduces to $\partial_r(rE_{u\phi}) = 0$. This means that there is a single non-trivial term at order $r^{-1}$ in the Einstein equation $E_{u\phi}$; it turns out to be the evolution equation (A.17) for angular momentum. Since in addition $E_{ru} = 0 = E_{u\phi}$, the second Bianchi identity above reduces to $\partial_r(rE_{uu}) = 0$. There is thus a single non-trivial term in $E_{uu}$, namely the evolution equation (A.18) for the mass aspect.

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
