# Peer review of "Radiative Asymptotic Symmetries of 3D Einstein--Maxwell Theory"

_SciPost Physics_

## Round 1 · Referee Report · Anonymous (Referee 1) · 2024-2-11

Strengths

1-Provides the first discussion of "news" in a 2+1 dimensional context
2-Einstein-Maxwell gravity in 3d hence provides a simple toy model that overcomes issues with other 3d toy models (radiation to scri, news, mass loss formula, etc.)
3-Authoritative discussion of asymptotic symmetries and associated algebras with high level of precision and generality

Weaknesses

1-Model does not allow for black holes in absence of cosmological constant and hence has limited scope

Report

I was impressed when reading this paper. While one could say "this could have been done in the 1990ies" it is a fact that it wasn't, and it takes both courage and endurance to embark on such an adventure. In my opinion, the authors did a great job at all levels: in the abstract and introduction, they made the case so clear that even outsiders to the field should be able to appreciate what this paper does, why it does it, and why the results are relevant for physics and of technical interest. In the body of the paper, the technical results are presented in sufficient detail to allow students and experts to follow the key developments while avoiding being too verbose. The language used is very clear and unpretentious. My only (minor) qualm is with section 6, which appears like an afterthought (and perhaps that is what it really is, and it is ok). I am sure that more could be said about (A)dS-Einstein-Maxwell in 3d, but I am fine with the authors' choice not to elaborate further on this story in the present paper. The outlook section 7 is also useful and enjoyable to read. However, from my perspective,there is a glaring omission: the authors do not mention a possible supersymmetric extension, which seems slightly odd.

I am happy to recommend the paper for publication in SciPost Phys., either in its present form or with the suggested inclusion on SUSY (see below), without further review.

Requested changes

I wouldn't say I request this change, but I strongly encourage the authors to add in the outlook section 7 another item in their itemized list addressing prospects of generalizing their analysis to supersymmetric (AdS)-Maxwell-Einstein in 3d.

---

## Round 1 · Referee Report · Anonymous (Referee 2) · 2024-2-16

Strengths

1 - Presentation and complete study of a concrete non-trivial radiative system in three-dimensional gravity with matter that mimics crucial features of four-dimensional radiative gravitational phase space.
2 - Comprehensive, detailed and interesting discussion of the various proposals for modified brackets for algebras of non-integrable charges.
3 - Provides a good technical background for future analyses in flat space holography and a formulation of celestial amplitudes in three dimensions.
4 - The quality of the writing and the clarity of the presentation make the reading really enjoyable and enriching.

Weaknesses

None

Report

This paper studies the asymptotic dynamics of three-dimensional Einstein gravity coupled to a Maxwell field, for both vanishing and non-vanishing values of the cosmological constant. Since the electromagnetic field describes a propagating degree of freedom in three dimensions, the combination of the two theories provides a nice toy model for radiative spacetimes in four-dimensional Einstein gravity, which has the good taste of being technically more tractable, but from which one can still draw important conclusions than can be applied to the more complicated higher-dimensional case.

For example, it is shown that the asymptotic charges are not conserved in (retarded) time, since Einstein-Maxwell dynamics prescribes flux-balance laws instead of conservation laws due to the electrodynamic radiative degree of freedom, which mimic the celebrated Bondi mass and angular momentum flux-balance equations in four dimensions. Moreover, due to the coupling with radiative fields, the asymptotic charges are found to be non-integrable. Since this non-integrability is essential in the presence of radiation, \textit{i.e.} it cannot be completely removed by a field-dependent transformation of the gauge parameters, several proposals for modified charge brackets have emerged, from the seminal work of Barnich and Troessaert to more recent approaches (\textit{e.g.} ref. [98]).

One of the key points of the paper is the application of the different proposals to the concrete and technically less complicated system presented above and the associated critical description of the virtues and the drawbacks of each definition, which is very interesting. This can shed new light on previous analyses of covariant phase space descriptions of radiative gravitational systems, and teach us much for future endeavours, namely on how to construct an ``ultimate'' version of the modified charge bracket that could consistently lift the standard Peierls/Dirac bracket to systems with non-conservative boundary conditions, hence non-conserved/non-integrable asymptotic charges.

The paper is very well written, the motivations are always clear and detailed, and the Authors have taken particular care to write the mathematical derivations in such a way that one can easily and fluently jump from one step to another. The results are scientifically sound and timely, considering the growing interest in flat space holography in the HEP community, for which one needs to understand how to encode Bondi flux-balance laws from an intrinsic boundary perspective. As it discusses a theory with non-trivial electrodynamic bulk scattering from the point of view of null infinity, this paper could also be considered as a first step towards the formulation of celestial amplitudes in three dimensions, where the ``holographic screen'' is no longer the null plane at infinity, but the one-dimensional celestial circle. In conclusion, I am happy to recommend this paper for publication in SciPost for the quality of the presentation, the relevance of the results to current research, and for the numerous avenues it can open for future investigations.

Requested changes

None

---

## Editorial Decision

resubmitted